# INDEQS: Informed Neural controlled Differential EQuationS

## Abstract

Neural Controlled Differential Equations (NCDE) provide a powerful continuous-time framework for forecasting time series, but standard graph-based extensions typically learn spatial structure purely from data, even in settings where a directed graph structure is known a priori. We introduce Informed Neural controlled Differential EQuationS (INDEQS), a modification to graph-based NCDE forecasting methods that incorporates prior knowledge of a directed graph at distinct architectural positions. INDEQS separates *inner* mixing of hidden states across graph nodes from *outer* mixing between vector field and control, and offers both a lightweight graph-constrained variant and a more expressive variant, learning additional graph connections from data via adaptive graph convolutions. To systematically study when graph informedness is beneficial in forecasting, we devise a continuous advection simulation on directed graphs, yielding synthetic spatio-temporal datasets with known ground-truth flow structure. We then evaluate INDEQS on two real-world tasks: river discharge forecasting on a hydrological network and traffic flow prediction on PeMS08. Across the synthetic and the river-discharge tasks, outer informedness consistently improves mean absolute error over an uninformed NCDE with comparable parameter count, particularly on larger graphs, while inner informedness offers a more parameter-efficient alternative when strict adherence to a known adjacency is desired. A comparison of discrete convolutional and continuous-time decoders further shows that continuous decoders yield better accuracy and greater temporal flexibility on real-world tasks. An implementation of INDEQS and the advection simulation is available at `https://anonymous.4open.science/r/indeqs_tmlr-588E`.

## 1 Introduction

Effect follows cause. When a problem is represented on a graph, the structure of that graph contains information on how causes at one spatial position are linked to effects at another, and learning physical dynamics from spatial time series data can often leverage this structural information. Schölkopf (2022) describes differential equations as the gold standard for understanding cause-effect structures and highlights how standard machine learning methods give limited treatment to time. Neural Differential Equations (NDEs) Chen et al. (2018) address this gap by learning a hidden state that evolves continuously in time. Neural Controlled Differential Equations (NCDEs) (Kidger et al., 2020) extend this further, updating the hidden state continuously from data arriving at different points in time, and Spatio-Temporal Graph NCDEs (STG-NCDEs) (Choi et al., 2022) additionally model spatial dependencies through learned node embeddings.

In physical systems with known directed graph structure such as river networks, road networks, and other advective systems, the underlying graph structure carries information on how causes propagate to effects. Existing graph-NCDE approaches learn the spatial structure from data, even when this prior is available.

We propose Informed Neural controlled Differential EQuationS (INDEQS), which inject prior information about an underlying directed graph into the vector field of a Neural Controlled Differential Equation (NCDE) to infer the future dynamics at the vertices. INDEQS comes in two variants — inner and outer informedness — that differ in where the graph structure enters the architecture. Intuitively, the inner position constrains where each node looks across the graph to assemble its update, while the outer position constrains how information from the control is distributed across nodes.

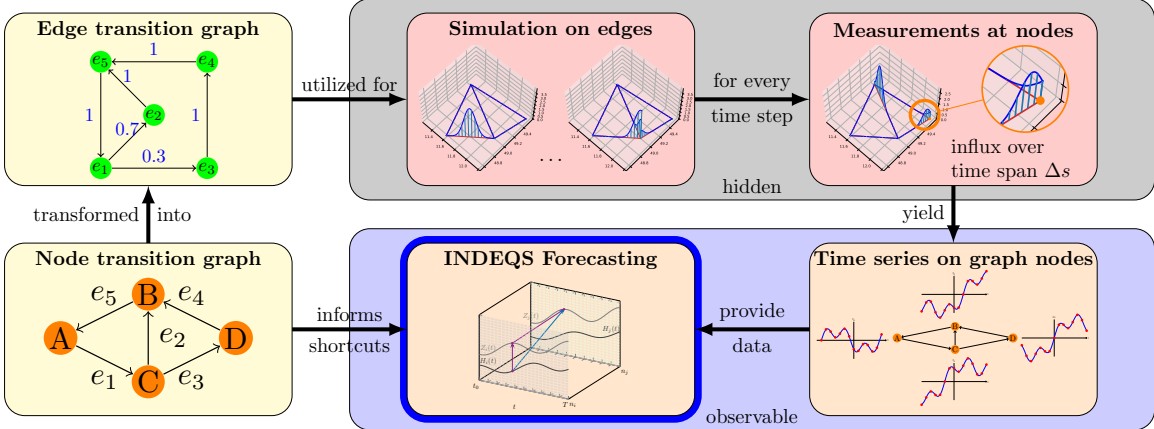

Figure 1: Overview: Starting from an edge transition graph with prior known structure, we simulate advection on the graph edges. Through measurements at vertices, one moves from a description as a continuum on edges to a node-wise time series description of the dynamic. These are used in INDEQS to forecast future state values. Graph information comes via the node transition graph, which can be transformed to the edge transition graph. Edges of the former become nodes in the latter. The first row shows the dynamics on an edge level, the second row on a node level, with the measurements being the bridge from edge representation to node representation. In a real-world application, one cannot observe the propagation along the edges continuously in space over time. One observes the system only at the measuring nodes. INDEQS allows supplementing time series data on nodes with graph information coming from prior knowledge of the dynamics of interest.

To study in a controlled experiment when graph informedness helps, we develop a continuous-time advection simulation on directed graphs, which yields synthetic datasets with known ground-truth flow structure. We then evaluate INDEQS on two real-world tasks: river discharge forecasting on a hydrological network and traffic flow prediction on PeMS08. We restrict our scope to static graphs and use continuous-time models as our primary comparisons; discrete-time baselines are included to situate INDEQS within the broader forecasting literature. Our **contributions** are:

- We introduce a modification to the graph NCDE method STG-NCDE, called INDEQS, that incorporates prior directed-graph information into the vector field at two architectural positions: inner (mixing across nodes' states) and outer (mixing between hidden state and control), yielding a parameter-efficient variant that strictly respects the known adjacency and a more expressive, data-adaptive variant.

- We propose a continuous-time advection simulation on directed graphs, which produces synthetic spatio-temporal datasets with known ground-truth flow structure and enables controlled study of when graph informedness benefits forecasting.

- Across a synthetic and a real-world benchmark (river discharge), outer informedness reduces MAE relative to an uninformed NCDE at matched parameter count, with the largest gains on larger graphs; inner informedness offers a parameter-efficient alternative that strictly respects the known adjacency.

Figure 1 shows an overview of the steps from continuous edge advection simulations to node measurements, and the incorporation of the underlying graph structure of the dynamic into the INDEQS forecasting method.

## 2 Related Work

Spatio-temporal forecasting encompasses a wide range of model families, differing in how they treat the spatial and temporal domains. Ju et al. (2024) give overviews for graph representation learning and Jin et al. (2024) give an overview of graph neural networks for time series forecasting. In Section A.13 we provide a tabular comparison with the most closely related Graph Neural Differential Equation methods to our work. In the following subsections, we outline different aspects of related works to outline the different

treatments of the temporal dimension (discrete versus continuous), and the discrete spatial dimension of graphs versus discretizations of continuous space. Moreover, we address composites of continuous time plus graphs, and conclude with continuous simulations and general applications.

## 2.1 Graph information in time-discrete forecasting models

Some time-discrete methods learn the graph structure from the given time series data, others rely on a prior given graph. Spatial-Temporal Graph Neural Networks (GNNs) (Wu et al., 2021; Rossi et al., 2020) inform Neural Networks (NNs) with graph information, by combining Message Passing NNs (Gilmer et al., 2017) with Recurrent Neural Networks (RNNs) like Long Short Term Memorys (LSTMs) (Hochreiter & Schmidhuber, 1997) or Gated Recurrent Units (GRUs) (Cho et al., 2014) leading to methods like Diffusion Convolutional Recurrent Neural Networks (DCRNNs) (Li et al., 2018), which depend on a prior known graph structure. Wu et al. (2019) combine Graph Convolutional Networkss (GCNs) (Kipf & Welling, 2017) and dilated convolutions (Yu & Koltun, 2016) for the temporal processing, leading to Graph WaveNet (GWaveNet), which has one variant that learns the graph structure from data and another that uses a prior given graph. Spatial-Temporal Graph Convolutional Networks (STGCNs) (Yu et al., 2018) use graph convolution in both the spatial and the temporal processing, also requiring the prior graph structure. Graph Multi-Attention Network (GMAN) (Zheng et al., 2020) uses attention (Kim et al., 2017; Bahdanau et al., 2014) for spatial and temporal processing and does not make use of a given graph topology, only learning graph connectivity from data. Attribute-Augmented Spatiotemporal Graph Convolutional Networks (AST-GCNs) (Zhu et al., 2021) combine graph convolutions on a fixed given graph and GRUs, along with additional node attribute information, while not performing graph adaptions based on data. Attention Based Spatial-Temporal Graph Convolutional Networkss (ASTGCNs) (Guo et al., 2019) use spatial and temporal attention, assuming a given graph structure and only dynamically reweigh edge connections from data. Xiao et al. (2025) use self-attention and learnable node-embeddings to capture graph dynamics and GRUs as the temporal processing.

## 2.2 Time-continuous models

The seminal work of NDEs (Chen et al., 2018), along with Funahashi & Nakamura (1993), allow continuous layer depth which can be arranged to reflect continuous time to align the data time dimension with the model's internal, virtual time dimension. To naturally adapt to later incoming data points, the notion of Controlled Differential Equations (CDEs) is used in NCDEs (Kidger, 2021), allowing learning path-dependent maps from path-space to path-space. The control path construction and their properties is explored in Morrill et al. (2021) and Morrill et al. (2022). Walker et al. (2024) introduce a training method to increase efficiency of NCDEs using rough path theory (Lyons et al., 2007). Rubanova et al. (2019) and Dupont et al. (2019) augmented Neural Ordinary Differential Equations (NODEs). Jhin et al. (2024) use attention on the output of an NCDE together with the data path. An overview over different NDE models on time series tasks is provided by Oh et al. (2025).

## 2.3 Graph time-continuous models

The continuous methods can be extended to allow treatment of time series on graphs. Poli et al. (2021) introduce spatial inductive bias as a node embedding into the vector field while generalizing the notion of NODEs to graphs. By evolving a latent space representation of the edge connections coupled with a latent state space of node features, Huang et al. (2021) proposed a method that allows modeling dynamic graph structures over time together with temporally varying node features. Hereby a GNN is used to compute the latent initial states for edges and nodes, respectively, based on the observed sequence of node attributes and adjacency matrix sequences, whereby the temporal dimension is treated discretely via temporal edges. These states are then evolved via a coupled ODE to predict future nodes and edges, via the likelihood of the states. Choi et al. (2022) consider time series on a graph by using a graph embedding to capture the spatial dependencies from data, without directly incorporating prior graph knowledge for traffic forecasting, as the graph connection between nodes is learned not given. InJiang et al. (2023a) the initial latent representation is first learned from initial observations at a point in time, then the continuous latent representation trajectory

is learned as the solution to an ODE, which incorporates features as additional inputs via a GNN. Luo et al. (2023) use a graph with temporal edges to represent a time series and extract the initialization of latent states via graph convolution together with a spectral convolution, and evolve future node and edge representations using second-order ODEs. Qin et al. (2026) propose Graph Neural Controlled Differential Equationss (GN-CDEs), a method to learn a dynamic graph embedding by creating graph paths in the controlled differential equation, allowing for time dependent graph structures, rather then having a static graph, the whole graph control path over time has to be given. Berndt et al. (2026) introduce permutation equivariance into GN-CDEs. Chen et al. (2024) use two adaptive graph convolutions with opposite signs in their spatial architecture. Liu et al. (2025) give an overview of graphs and GNNs in NDEs.

### 2.4 Graphs as approximations of spatial continuum

Continuous methods exist that can be seen as a continuous version of GNNs, but they do not treat time series: Xhonneux et al. (2020) continuously diffuse along the spatial graph structure. GRAND (Chamberlain et al., 2021) is a method that treats a GNN as a discretization of an underlying diffusion process. CGNN (Xhonneux et al., 2020) generalize GNNs by seeing them as a specific discretization scheme. Continuous Graph Flow (Deng et al., 2019) are Ordinary Differential Equations (ODEs) that operate on a graph structure and Cranmer et al. (2020) distill symbolic representations of a learned deep model by introducing strong inductive bias via a GNN.

### 2.5 Simulations, applications, and other spatial information strategies

Chapman & Mesbahi (2011) formulated advection on graphs, Zang & Wang (2020) simulate network dynamics based on heat diffusion, mutualistic interaction of species and gene regulatory dynamics, and Arndt et al. (2025) use Partial Differential Equations (PDEs) to generate synthetic data for spatio-temporal graph forecasting. All three do not consider advection along graph edges. Maddix et al. (2022) describe a similar simulation to ours, with continuous time evolution at the nodes, but do not consider a continuous edge domain between nodes. Kirschstein & Sun (2024) investigate graph topology information for GNNs on a river discharge problem. Kratzert et al. (2018) forecast rainfall runoffs using LSTMs, however use additional weather data as features and predict gauge-wise, different to our setting. Jiang et al. (2023b) and Liu & Zhang (2024) give an overview of GNN for traffic forecasting. Apart from the discrete-time methods for traffic forecasting (Fang et al., 2021; Liu et al., 2023; Li et al., 2018; Lan et al., 2022; Xiao et al., 2025; Wu et al., 2019; Yu & Koltun, 2016; Yu et al., 2018; Zheng et al., 2020; Zhu et al., 2021; Guo et al., 2019; Shao et al., 2022; Weng et al., 2023), continuous-time methods were also used for this domain (Poli et al., 2021; Choi et al., 2022; Choi & Park, 2023). Kosma et al. (2023) use NODEs for epidemic spreading on graphs and Verma et al. (2024) use NODEs for climate and weather forecasting on discretized continuous space. Other attempts of incorporating prior spatial knowledge into NN architecture include Physics Informed Neural Networks (PINNS) (Raissi et al., 2019), which enforce a prior defined PDE but do not act on graphs, as they avail oneself of a discretized continuous description of the spatial domain.

## 3 Background

NDEs model the evolution of a hidden state as the solution to a differential equation parameterized by a NN (Chen et al., 2018; Kidger, 2021). By treating time as continuous, NDEs process irregularly sampled observations naturally and admit memory-efficient training. This section recapitulates NODEs and the variants of NDEs that INDEQS builds on: NCDEs and their graph-based extensions.

### 3.1 Neural Ordinary Differential Equations

NODEs Chen et al. (2018) are time-continuous models that describe the evolution of a hidden state $y \in \mathbb{R}^{d_y}$ via a differential equation up to time $T > 0$

$$y(0) = y_0, \quad \frac{\mathrm{d}y(t)}{\mathrm{d}t} = f_\theta(t, y(t)), \quad t \in [0, T],$$

where the vector field on the right-hand side is a neural network $f_\theta : [0, T] \times \mathbb{R}^{d_y} \to \mathbb{R}^{d_y}$, with parameters $\theta$ — rendering the differential equation "neural". Lipschitz continuity of $f_\theta$ ensures the existence and uniqueness of a solution (Kidger, 2021). For some data input $x_0 \in \mathbb{R}^{d_x-1}$ at time $t_0$, an affine mapping $m_\theta$ is applied to obtain the initial value for the differential equation $y_0 = m_\theta((t_0, x_0)) \in \mathbb{R}^{d_y}$, and thereby augmenting the NDE (Dupont et al., 2019). After evolving through virtual time up to $T$, the final state $y(T)$ serves as input to another affine layer $l_\theta$ to obtain the output of the model. NODEs can be considered the continuous-time limit of residual networks (He et al., 2016) and can also be formulated in integral form via

$$y(0) = y_0, \quad y(T) = y(0) + \int_0^T f_\theta(s, y(s)) \, \mathrm{d}s.$$

## 3.2 Neural Controlled Differential Equations

For modeling multivariate time series with observations $\{(t_i, x_i)\}_{i=0}^n$, one can turn to NCDEs (Kidger et al., 2020), which incorporate new observations continuously over time rather than only at initialization. To do so, the discrete observations are interpolated into a continuous path $x : [0, T] \to \mathbb{R}^{d_x}$ that acts as the control for the differential equation. An NCDE driven by $x$ is then defined by

$$y(0) = y_0, \quad y(T) = y(0) + \int_0^T f_\theta(y(s)) \, \mathrm{d}x(s),$$

with solution $y : [0, T] \to \mathbb{R}^{d_y}$, where $f_\theta : \mathbb{R}^{d_y} \to \mathbb{R}^{d_y \times d_x}$ is a neural network assumed to be Lipschitz continuous and the integral is taken in the Riemann–Stieltjes sense against the control path $x$, wherein the integrand $f_\theta(y(t)) \, \mathrm{d}x(t)$ is a matrix-vector product. This control path $x$ allows a continuous information inflow from data across $[0, T]$, in contrast to the NODE setting, where information enters only at $t = 0$. NCDEs can be considered operators that map from path space to path space. The initial state $y_0$ is obtained via a learnable function $n_\gamma : \mathbb{R}^{d_x} \to \mathbb{R}^{d_y}, x(0) \mapsto n_\gamma(x(0)) = y_0$. NCDEs can be seen as the continuous version of RNNs, as later data is included as one progresses along the time series. The output is obtained through application of an affine layer $l_\theta$ via $o_t = l_\theta(y_t)$. For a differentiable path of bounded variation, reduction of the NCDE to an NODE is possible to attain

$$y(0) = y_0, \quad y(T) = y(0) + \int_0^T f_\theta(y(s)) \frac{\mathrm{d}x(s)}{\mathrm{d}s} \, \mathrm{d}s,$$

which allows the use of the training techniques of NODEs, with the integral now being a Riemann integral and the integrand $f_\theta(y(t)) \frac{\mathrm{d}x(t)}{\mathrm{d}t}$ again refering to a matrix-vector product.

## 3.3 Graph Neural Controlled Differential Equations

We write in the following $\{\mathcal{G}_{t_i} \triangleq (\mathcal{V}, \mathcal{E}, \mathbf{X}_{t_i})\}_{i=0}^n$ for a multivariate time series $\mathbf{X}_{t_0}, \ldots, \mathbf{X}_{t_n} \in \mathbb{R}^{|\mathcal{V}| \times d_x}$ observed at times $t_0, \ldots, t_n$ on a directed graph $\mathcal{G}$ with vertices $v \in \mathcal{V}$ and edges $e \in \mathcal{E}$. To model this setting, Choi et al. (2022) stack the state vectors of NCDEs to matrices in order to represent the additional graph dimensionality $|\mathcal{V}|$, and use $d_x, d_h, d_z$ to represent the dimensionality of a single node for the control and the two hidden states, respectively. This leads to (see appendix A.2 for dimensional details)

$$\boldsymbol{H}(T) = \boldsymbol{H}(0) + \int_0^T \boldsymbol{f_\theta}\Big(\boldsymbol{H}(t)\Big) \frac{d\boldsymbol{X}(t)}{dt} dt, \tag{1}$$

wherein the hidden state $\boldsymbol{H}(t) \in \mathbb{R}^{|\mathcal{V}| \times d_h}$ is continuously updated in time, starting from $\boldsymbol{H}(0) = \boldsymbol{H}_0 \in \mathbb{R}^{|\mathcal{V}| \times d_h}$, $\boldsymbol{X}(t) \in \mathbb{R}^{|\mathcal{V}| \times d_x}$ is the control and $\boldsymbol{f_\theta} : \mathbb{R}^{|\mathcal{V}| \times d_h} \to \mathbb{R}^{|\mathcal{V}| \times d_h \times d_x}$ is the guiding vector field. The integrand constitutes a matrix-vector product batched across nodes. Jhin et al. (2024) and Kidger (2021, p.66) show that NCDEs can be coupled to use the encoded hidden path $\boldsymbol{H}$ as a control for a second NCDE with hidden state $\boldsymbol{Z}(t) \in \mathbb{R}^{|\mathcal{V}| \times d_z}$, which reduces to the NODE

$$\boldsymbol{Z}(T) = \boldsymbol{Z}(0) + \int_0^T \boldsymbol{g_\gamma}\Big(\boldsymbol{Z}(t)\Big) \frac{d\boldsymbol{H}(t)}{dt} dt, \tag{2}$$

with initial value $\boldsymbol{Z}(0) = \boldsymbol{Z}_0 \in \mathbb{R}^{|\mathcal{V}| \times d_z}$ and vector field $\boldsymbol{g_\gamma} : \mathbb{R}^{|\mathcal{V}| \times d_z} \to \mathbb{R}^{|\mathcal{V}| \times d_z \times d_h}$. Again the integrand is a matrix-vecor product batched across nodes. The coupled NODE describing the dynamics takes the form (intermediary steps can be found in A.3)

$$\frac{d}{dt} \begin{bmatrix} \boldsymbol{Z}(t) \\ \boldsymbol{H}(t) \end{bmatrix} = \begin{bmatrix} \boldsymbol{g_\gamma}\Big(\boldsymbol{Z}(t)\Big) \boldsymbol{f_\theta}\Big(\boldsymbol{H}(t)\Big) \frac{d\boldsymbol{X}(t)}{dt} \\ \boldsymbol{f_\theta}\Big(\boldsymbol{H}(t)\Big) \frac{d\boldsymbol{X}(t)}{dt} \end{bmatrix}. \tag{3}$$

One now constrains $\boldsymbol{f_\theta}$ to process information for every node separately to only capture the temporal dependencies individually, without mixing information across nodes, whereas $\boldsymbol{g_\gamma}$ learns the spatial dependencies.

For forecasting, the final state $\boldsymbol{Z}(T)$ encodes the information gathered from time 0 up to $T$ and is used as input to a final convolutional layer to make the final predictions $\{\hat{\mathbf{Y}}_{t_i}\}_{i=0}^n$ with $T < t_i \leq T + \tau$ at time $T$ for the next $M$ time steps up to time $T + \tau$. One can solve and backpropagate through the NCDE given a loss function $\mathcal{L}(\{\hat{\mathbf{Y}}_{t_i}\}_{i=0}^n, \{\mathbf{Y}_{t_i}\}_{i=0}^n)$ and optimize the parameters $\boldsymbol{\theta}$ and $\boldsymbol{\gamma}$.

# 4 Informed Graph Neural Controlled Differential Equations (INDEQS)

We present Informed Neural controlled Differential EQuationS (INDEQS) to incorporate prior directed graph structure into Graph Neural Controlled Differential Equations (GNCDEs), which act as an encoder of the features in the context window. The encoding is then used by the decoder heads to forecast future time points. We define two variations of INDEQS: inner and outer informedness. NCDEs in contrast to NODEs offer the unique possibility to use the interaction between vector field and control, to introduce connections between the nodes' hidden states in a different way. Instead of only mixing information between the nodes' states via the vector field itself, the outer informedness introduces direct spatial shortcuts between control and vector field, leading to connections between control (the later incoming data) and the nodes' states. To highlight the differences in graph informing positions within the architecture of an NCDE, we distinguish between the inner mapping

$$\boldsymbol{g_\gamma}(\cdot) : \mathbb{R}^{d_{\boldsymbol{Z}}} \to \mathbb{R}^{d_{\boldsymbol{Z}} \times d_h} \tag{4}$$

and the outer mapping

$$\boldsymbol{g_\gamma}(\mathbf{Z}(t)) \bullet \cdot : \mathbb{R}^{d_{\boldsymbol{H}}} \to \mathbb{R}^{d_{\boldsymbol{Z}}}, \tag{5}$$

that the vector field $\boldsymbol{g_\gamma}$ in Equation (2) describes, using $\bullet$ to emphasize the tensor product, where $d_{\boldsymbol{Z}} = |\mathcal{V}| \times d_z$ and $d_{\boldsymbol{H}} = |\mathcal{V}| \times d_h$. Whereas Equation (4) describes the inner workings of the update of the hidden state $\mathbf{Z}$ on itself, the tensor multiplication in Equation (5) describes the direct connection between control (and thereby the data) to the hidden state. Naturally, both happens simultaneously. The function in Equation (4) describes how a hidden state entry at a node has to encode "where to look in the control" in its $d_h$-dimensional state based on the entries of other nodes' hidden states and itself. In contrast, the function in Equation (5) describes for a fixed $\boldsymbol{g_\gamma}$ the situation in which one "knows where to look" to read out the right hidden state update information from the control. In STG-NCDE (Choi et al., 2022) the outer mapping is only used node-wise for a node's $d_h$-dimensional state and the node's $d_x$-dimensional control, due to the batched matrix multiplication across nodes. For that reason we introduce an additional mixing mechanism at this outer position in INDEQS$_{\text{outer}}$ to allow mixing of a hidden state at node $i$ and the control at node $j$ for adjacent graph nodes in a prior given directed graph. Figure 2 schematically illustrates the differences in the connections.

## 4.1 Outer informed with inner node embedding: INDEQS$_{\text{outer}}$

To define $INDEQS_{outer}$, we use the prior known transposed vertex transition matrix $\boldsymbol{A}_{\mathcal{V}}^{outer}$ to inform our NCDE at the outer postion via

$$\boldsymbol{Z}(T) = \boldsymbol{Z}(0) + \int_0^T \boldsymbol{g_\gamma}\Big(\boldsymbol{Z}(t)\Big) \mathbf{A}_{\mathcal{V}}^{outer} \frac{d\boldsymbol{H}(t)}{dt} dt, \tag{6}$$

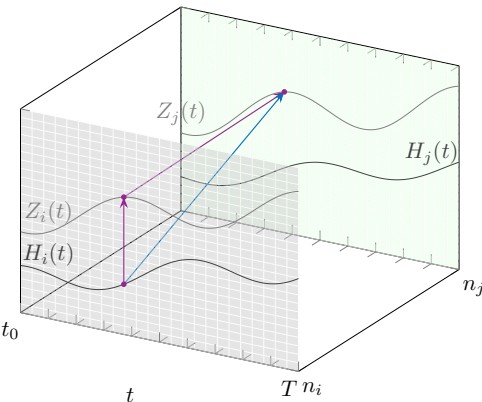

Figure 2: Difference in information transport between hidden state $\mathbf{H}_i$ at node $n_i$ and hidden state $\mathbf{Z}_j$ at node $n_j$, occurring at every point in time $t$. Connection with via the outer mapping (blue) leads to a direct connection versus connecting nodes via the inner mapping (violet).

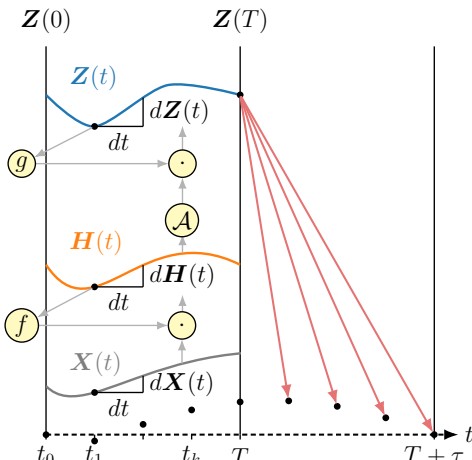

Figure 3: NCDE encoder and convolutional decoder. **Encoding** (left half): Up to time $T$, the data points at the bottom are interpolated to obtain a continuous path $\boldsymbol{X}$, whose derivative is then multiplied with $f$, to obtain $d\boldsymbol{H}$. $\mathcal{A}$ allows now direct mixing – at the *outer* position – of the nodes' control $d\boldsymbol{H}$, depending on $d\boldsymbol{H}$. Only after occurs the multiplication with $g$ to obtain the final update $d\boldsymbol{Z}$. A mixing at the *inner* position within $g$ can only be dependent on $\boldsymbol{Z}$. Multiplication here means tensor multiplication, and the 1-dimensionally depicted data and the paths, are multidimensional with multiple nodes and multiple dimensions per node. **Decoding** (right half): Based on the last hidden state $\boldsymbol{Z}(T)$, the convolution head (red arrows) directly makes all future prediction at only at predefined times.

where "outer" is referring to the position of the transposed node adjacency matrix. In the integrand, $\boldsymbol{g}_\theta \boldsymbol{A}_{\mathcal{V}}^{\text{outer}}$ is a batched matrix mutliplication across the nodes and $\boldsymbol{A}_{\mathcal{V}}^{\text{outer}} \frac{d\boldsymbol{H}(t)}{dt}$ is a matrix multiplication. The internal mapping is given by

$$\boldsymbol{g}_{\boldsymbol{\gamma}}(\boldsymbol{Z}(t)) = g_3 \circ g_2 \circ g_1(\boldsymbol{Z}(t)),$$

with $g_1 : \mathbb{R}^{d_{\boldsymbol{Z}}} \to \mathbb{R}^{d_{\boldsymbol{Z}}}$, $g_3 : \mathbb{R}^{d_{\boldsymbol{Z}}} \to \mathbb{R}^{d_{\boldsymbol{Z}} \times d_{\boldsymbol{H}}}$ being fully connected layers with ReLU and hyperbolic tangent activation. INDEQS$_{\text{outer}}$ follows the parameterization of the inner $g_2$ via

$$g_2(B) = (\boldsymbol{I} + \phi(\sigma(\boldsymbol{E} \cdot \boldsymbol{E}^T)))BW, \tag{7}$$

as proposed in Choi et al. (2022), where $W \in \mathbb{R}^{d_{\boldsymbol{Z}}} \times \mathbb{R}^{d_{\boldsymbol{Z}}}$ is a trainable weight transformation matrix, that reweighs each nodes's high dimensional representation the same way, $\boldsymbol{E} \in \mathbb{R}^{|\mathcal{V}| \times C}$ is a learnable node embedding with embedding dimension $C \in \mathbb{N}$, $\phi$ is a row wise softmax activation and $\sigma$ is meant to represent elementwise ReLU. Equation (7) is a graph convolutional operation (Kipf & Welling, 2017) with an adaptive, learnable adjacency matrix. Setting $\boldsymbol{A}_{\mathcal{V}}^{outer}$ to the identity fully recovers the setting of Choi et al. (2022). Throughout our experiments, this setting serves as the baseline, which we denote as *STG-NCDE*, following Choi et al. (2022).

## 4.2 Inner informed with outer identity: INDEQS$_{\text{inner}}$

To define *INDEQS$_{inner}$* we set $\boldsymbol{A}_{\mathcal{V}}^{outer}$ to the identity matrix but replace the node embedding in Equation (7) with

$$g_2(B) = (\boldsymbol{A}_{\mathcal{V}}^{inner})B,$$

directly incorporating the transposed vertex transition matrix $\boldsymbol{A}_{\mathcal{V}}^{inner}$ at the inner position of the internal mapping. INDEQS$_{\text{inner}}$ has fewer parameters than STG-NCDE and INDEQS$_{\text{outer}}$, as the adjacency matrix is hardcoded without any learnable parameters, replacing the node embedding.

INDEQS$_{\text{inner}}$ constrains the flow of information to allow only communication between nodes which are connected via the adjacency matrix. This way one can make sure, that causally unrelated nodes are not allowed to communicate directly. For INDEQS$_{\text{outer}}$ the model has graph structure information, but is not constrained, as the model still is able to learn other patterns apart from the given adjacency connections. In the setting of STG-NCDE, one allows connections between all nodes and these connections are learned. In addition, we would like to point out that the configuration with both the outer and inner mechanisms set to an identity matrix, would keep each node information completely separate from each other, only processing the nodes' own state temporally, which would be equal to having $n-$many $1-$node models processed separately. The outer position of INDEQS can be seen analogous to a shortcut connection in a Residual Neural Network (ResNet) (He et al., 2016), only between spatial nodes and not in the virtual time dimension.

The encoding is visually depicted in Figure 3, illustrating the different information positions with $\mathcal{A}$ at the outer position and $g$ containing the inner graph connectivity position.

## 4.3 Higher order adjacencies

A predecessor of node $m$ is a node $n$ such that there exists a directed edge from $n$ to $m$. In pursuit of integrating information from the predecessors of predecessors and higher degrees into our NCDEs, we replace $\boldsymbol{A}_{\mathcal{V}}$ in INDEQS$_{\text{outer}}$ and INDEQS$_{\text{inner}}$ by the sum over the $k$-th power adjacency matrices

$$\boldsymbol{A}_{\mathcal{V}}(k) = \sum_{i=0}^{k} \boldsymbol{A}_{\mathcal{V}}^i, \tag{8}$$

where $k$ can be any integer less than or equal to the longest possible path length in the given directed graph. When informing our NCDE with adjacencies up to the $l$-th power via Equation (8), we refer to this configuration as INDEQS$_{\text{outer}}(k = l)$.

### 4.4 Continuous decoders

Instead of predicting future time steps with the default convolutional decoder, one can also continuously evolve the hidden states to be able to obtain predictions at continuous times via NODE based decoders that we outline in the appendix Section A.7.

## 5 Simulating Advection on Graph Edges

Continuous advection on a graph is most easily understood by following pulses as they travel along edges, fork at nodes according to predefined split ratios, and merge additively where multiple edges meet. One exemplary Gaussian pulse being advected on a 4-node graph is depicted in Figure 4. In the following, we

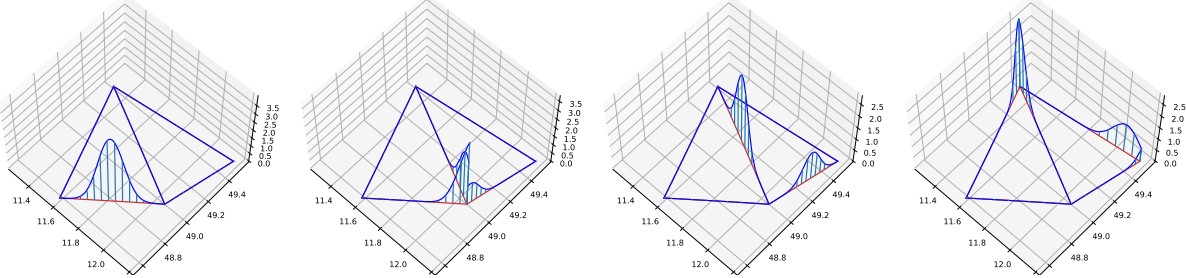

Figure 4: Advection of an initial Gaussian pulse on a 4 node graph with 5 edges over time

present the mathematical details behind the simulation of advection on graph edges.

For a controlled experimental setting, we propose a simulation for linear advection along graph edges. In contrast to Chapman & Mesbahi (2011), where the quantities on the vertices are updated discretely in time, we advect quantities that move continuously in time along the edges of a graph and fork or merge at vertices. For the advection simulation, we consider a directed graph $\mathcal{G}$ with a set of vertices $\mathcal{V}$ and edges $\mathcal{E}$. Every edge $e \in \mathcal{E}$ has a time-static continuous domain $\Omega_e = \{x \in \mathbb{R} \mid 0 \leq x < \delta(e)\}$ and a function space $V_e = \{y_e(x,t) \mid \Omega(e) \times \mathcal{T} \to \mathbb{R}^d\}$ attached to it, where $\delta(e)$ is the length of edge $e$ and $t \in \mathcal{T} = [0,T]$ is a point in time. We first focus on the description of the advection on the interior of the edges without considering transitions between edges: Given an initial state of the dynamic system it evolves according to an advection differential equation

$$\frac{\partial \mathbf{y}}{\partial t} + \nabla \cdot (\mathbf{y} \odot \mathbf{v}) = 0, \quad \mathbf{y}(\,\cdot\,,0) = \mathbf{y}_0 \tag{9}$$

with $\mathbf{v} = (v_1, \ldots, v_{|\mathcal{E}|})^\intercal$ being the vector of scalar velocity fields $v_e : \Omega_e \to \mathbb{R}$, with $e \in \{1, \ldots, |\mathcal{E}|\}$, that advect the quantity $\mathbf{y} = (y_{e_1}, \ldots, y_{e_{|\mathcal{E}|}})^\intercal$ through space over time. The divergence is intended row-wise and $\odot$ signifies element-wise multiplication. We restrict the entries of $\mathbf{y}$ to positive values, so that the flow direction is always the direction of the directed edge. We assume that the velocity field is constant. In the case of a one-dimensional quantity and velocity field along the edges, this leads to $\frac{\partial y_e}{\partial t} = -v_e \frac{\partial y_e}{\partial x}$, which has the analytical solution $y_e(x,t) = y_e^0(x - vt)$, with initial condition $y_e^0 := y_e(x,0)$, for all $x \in \Omega(e)$, for all $e \in \mathcal{E}$ and $v_e t \leq x \leq \delta(e)$. With this description of the dynamic, $y_e$ for $e \in \mathcal{E}$ is not required to be differentiable. Now, we describe the dynamic on the transition between adjacent edges. For $x < v_e t$ one would look back "beyond the edge".

We therefore resort to a matrix formulation to represent the edge transitions of $y$:

$$\mathbf{y}(\mathbf{x},t) = \left. \left(\mathbf{A}_\mathcal{E}\mathbf{y}(\mathbf{x},0)\right)\right|_{\mathbf{x}=(\delta(\mathbf{e})-v_e t \; \mathbf{1}_{|\mathcal{E}|})}, \quad x < v_e t, \tag{10}$$

where $t < \min_{e \in \mathcal{E}} \delta(e)/v_e$ has to hold for every edge to not look further back than the previous edge and where $\mathbf{A}_\mathcal{E}$ is a directed edge transition matrix, signifying to which edge a quantity transitions when the quantity reaches the end of an edge. $\mathbf{A}_\mathcal{E}$ acts as a function operator on the vector of functions $\mathbf{y}_0 =$

$(y_1^0, \ldots, y_{|\mathcal{E}|}^0)^\intercal$, and $\mathbf{x} = (x_1, \ldots, x_{|\mathcal{E}|})^\intercal$ is a collection vector of spatial coordinates on the different edges and $\mathbf{1} = (1, \ldots, 1)^\intercal \in \mathbb{R}^{|\mathcal{E}|}$. The entries $a_{ij}$ of $\mathbf{A}_\mathcal{E}$ are defined as

$$a_{ij} = \begin{cases} p_{ij} \in (0,1], & \text{if } e_i \text{ follows on } e_j \\ 0, & \text{else} \end{cases},$$

with $p_{ij}$ being the proportion of the quantity on $e_i$ transported to $e_j$ and $\sum_{i \in \mathcal{E}} p_{ij} = 1$ to enforce conservation. The matrix $\mathbf{A}_\mathcal{E}$ can be obtained from the the vertex adjacency matrix $\mathbf{A}_\mathcal{V}$ (details in A.4). We then iteratively advect with a recurrence equation derived from Equation (10) to derive

$$\mathbf{y}(\mathbf{x}, t + \Delta t) = \left( \mathbf{A}_\mathcal{E}(\mathbf{y}(\mathbf{x}, t)) \right)\Big|_{\mathbf{x} = (\delta(\mathbf{e}) - v\Delta t \; \mathbf{1}_{|\mathcal{E}|})},$$

which can be used for simulating data.

# 6 Experiments

We begin by describing the generation of the simulated data, followed by the description of the river discharge dataset and the traffic flow forecasting dataset.

## 6.1 Description of datasets

### 6.1.1 Advection simulation data

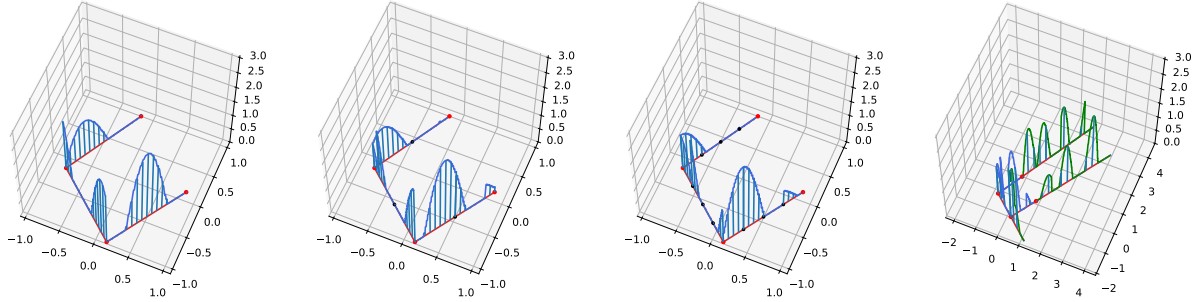

Figure 5: Advection on graph edges for a 4-node graph for 3 time steps and increasing node resolution. The additional intermediate nodes are depicted as black dots. The last image additionally shows a zoomed out view of another time step with future masses along the virtual edges (green) where masses enter at the top right edges and leave the graph on the bottom edge.

For the experiments on graph advection data, we first randomly sample graphs with number of nodes being $4, 8, 16, 32, 64$ and $128$, respectively, and hold these constant for the advection simulation along the edges. The (directed) graphs represent randomly generated Growing Network with Redirection (GNR) graphs (Krapivsky & Redner, 2001) (see A.5 for details), resembling the graph structure of a river network

$$y_0(x) = \max \left( 0, a_0 + \sum_{k=1}^{K} a_k \cos(2\pi kx/L) + b_k \sin(2\pi kx/L) \right), \tag{11}$$

where $K$ is a small random positive integer uniformly drawn from $\{2, \ldots, 20\}$, $a_0 = 0.2$, and the coefficients $a_k, b_k$ are independent Gaussian variables with decaying variance $k^{-\alpha}$ with $\alpha = 5$ (i.e. higher modes are damped), so that each edge carries a smooth, spatially correlated random profile with most power in low spatial frequencies.

We simulate the advection dynamic along the edges 2000 times with different initial conditions, for each of the graphs, and use $\Delta t = 8$ minutes to evolve for 24 steps into the future. To have enough mass influx

into the graph system for future steps, virtual edges are initialized accordingly (see Figure 5). To obtain data measurements at the nodes, the quantity that passed through the vertices in a given time span is aggregated. We take measurements at the nodes for $\Delta t = 8$ and acquire time series on the graph nodes $\{\mathcal{G}_{t_i} \triangleq (\mathcal{V}, \mathcal{E}, \mathbf{Y_{t_i}})\}_{i=0}^n$ that are used for training. The edges are of length 64 and the quantities move at speed $v = 1$. The task is to predict the quantities for the next 12 time points given the previous 12 time points, such that the previous nodes provide enough information to be able to infer the quantities at the subsequent nodes, which renders an adjacency matrix power of 1 the reasonable choice. We train on 80% of the data, use 10 % for validation and 10 % as a holdout test set. We train, validate and test on the non-source nodes, i.e. the nodes that have preceding nodes with measurements that would allow perfect prediction. For testing the stability of the methods for different spatial node resolutions $k$, we divide each edge into $k$ intervals, leading to $k - 1$ additional intermediate nodes, where additional measurements are taken. The graph adjacency matrix is rewired to reflect these additional nodes. Apart from the resolution, the graph structure, the initialization and the advective behavior stay the same. The different node resolutions are depicted in Figure 5. The code of the simulation and the generated data can be found in the code repository (`https://anonymous.4open.science/r/indeqs_tmlr-588E`).

### 6.1.2 River discharge data

In our real-world data experiments, we use runoff data from "The Global Runoff Data Centre, 56068 Koblenz, Germany". It includes daily measurements of mean discharges of the river Weser, recorded starting 1 Jul 1820 up till 25 Jul 2023 for at least one station. We selected the time span from January 6, 1998, to October 31, 2016 (6874 days), as it provided the longest possible overlapping period during which measurements were available for all stations. We excluded two stations because they contained no information on connected stations. The unit of measure for the discharge data is cubic meters per second $(m^3/s)$. This dataset provides valuable insights into the hydrological dynamics of the Weser River over an extended period. The dataset for the experiments features 45 nodes and 44 edges. We selected the Weser River network, as this river basin has a comparable size to our simulation experiments and provides information of the next downstream stations, allowing to create an adjacency matrix for informedness. The data can be found in the code repository. The task is to predict future discharges of the next 5 days given a context of the last 7 days, leading to 572 complete, disjoint time series fragments. For the training, we use a chronological 80/10/10 split for training, validation and test set, respectively. We use a stride of 1 to augment the number of samples to obtain 5489, 676 and 676 samples for training, validation and test set.

### 6.1.3 Traffic flow data

The PeMS08 dataset comprises district 8 of the the Caltrans Performance Measurement System (PeMS). It captures traffic flow measurements across the California highway system at a frequency of 5 minutes (288 measurements per day), spanning from 01 Jan 2016 to 31 Aug 2016. The monitored network consists of 170 nodes and 295 edges, providing a detailed representation of the sensor network in the San Bernardino area. To leverage the spatial dependencies, we construct an informed adjacency matrix $A \in \mathbb{R}^{N \times N}$ using a thresholded Gaussian kernel (Shuman et al., 2013) based on the physical distances $d_{i,j}$ between sensors:

$$w_{i,j} = \exp\left(-\frac{d_{i,j}^2}{\sigma^2}\right) \tag{12}$$

where $\sigma$ is the standard deviation of all recorded distances. We apply a threshold $r = 0.01$ such that $A_{i,j} = w_{i,j}$ if $w_{i,j} \geq r$, and 0 otherwise. This high-resolution temporal data enables in-depth studies of traffic fluctuations and dynamics within the network. Similar to the hydrological experiments, the task is to predict future traffic flow of the next hour given a context of the last hour, leading to 445 complete, disjoint time series fragments. For the training, we use a chronological 60/20/20 split for training, validation and test set, respectively. We use a stride of 1 to augment the number of samples to obtain 14,271, 1,784, and 1,784 samples, respectively, allowing for a precise analysis of the traffic patterns over the monitored period.

### 6.2 Results

We demonstrate the performance of INDEQS on the advection simulation data and a river discharge forecasting tasks on a river network while only using the endogenous discharges and no exogenous features, as the natural phenomenon of water flow exhibits an advective behavior, conserving masses moving steadily over time and we observe benefits of graph information for our advection simulation data. We also expect the river flows to behave conservatively to a degree and be more predictable than tasks solely involving human-interactions like traffic flow forecasting, which we will also study as the final task.

To train INDEQS we use hermite cubic splines with backward differences (Morrill et al., 2021) to construct differentiable paths $\mathbf{X}(t)$ from the given data points and *RK4*, a 4th order Runge-Kutta method (Runge, 1895; Kutta, 1901) to solve the differential equations. We train for 200 epochs for the advection simulation experiments, the river discharge and traffic flow problem and select the model that attained the best validation loss over the whole training period. For the simulation data, the discharge and traffic flow predictions, we train INDEQS in both variations and STG-NCDE with 5 random seeds.

#### 6.2.1 Advection Experiment

For the advection simulation experiment, we compare graph informed $\text{INDEQS}_{\text{outer}}$ and $\text{INDEQS}_{\text{inner}}$ to STG-NCDE. We observe in Figure 6 that the informed $\text{INDEQS}_{\text{outer}}$ (orange) is attaining a lower MAE compared to the uninformed STG-NCDE (blue) method for all graph sizes for spatial node resolution of 1 (see Figure 6 (a)) and 4 (see Figure 6 (c)), and also having lower MAE for spatial node resolution 2 for all considered graph sizes, except for a graph of size 8, where the performance is almost on par (see Figure 6 (b)). For a spatial resolution of 1 and 2, $\text{INDEQS}_{\text{outer}}$ exhibits a relatively stable MAE level over the node numbers, while the MAE of STG-NCDE is increasing with increasing graph sizes and then plateauing for 64 nodes and 128 nodes. For spatial resolution of 2, STG-NCDE exhibits increasing MAE for higher node numbers, a pattern observed also for spatial resolution of 4. $\text{INDEQS}_{\text{inner}}$ (green) has higher MAE than STG-NCDE for low node numbers, however exhibits lower MAE than STG-NCDE starting for higher node numbers. The turning point of this behavior increases with spatial resolution. All methods degenerate with increasing spatial resolution, while $\text{INDEQS}_{\text{inner}}$'s MAE increases most sharply for the 4-,8- and 16-node compared to $\text{INDEQS}_{\text{outer}}$ (see Figure 7). The results suggest that $\text{INDEQS}_{\text{outer}}$ is more stable across graph size and spatial resolution. All results are also provided in Table 4.

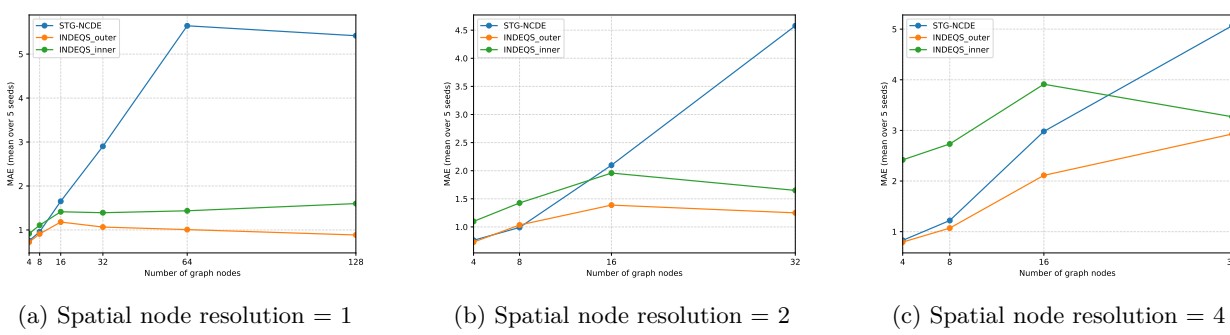

(a) Spatial node resolution = 1          (b) Spatial node resolution = 2          (c) Spatial node resolution = 4

Figure 6: MAE over 5 seed runs over graphs with increasing number of nodes for the **Advection Experiment** with fixed spatial node resolution.

#### 6.2.2 River discharges

For the river discharge experiment, we compare INDEQS to the uninformed STG-NCDE (Choi et al., 2022), the continuous graph informed Graph Convolutional ODE with Gated Recurrent Units (GCDE-GRU) (Poli et al., 2021) and the continuous graph informed STGODE (Fang et al., 2021). To contextualize the performance of our continuous model, we include comparisons with informed discrete-time recurrent model DCRNN (Li et al., 2018), and informed discrete-time convolution-based GWaveNet (Wu et al., 2019).

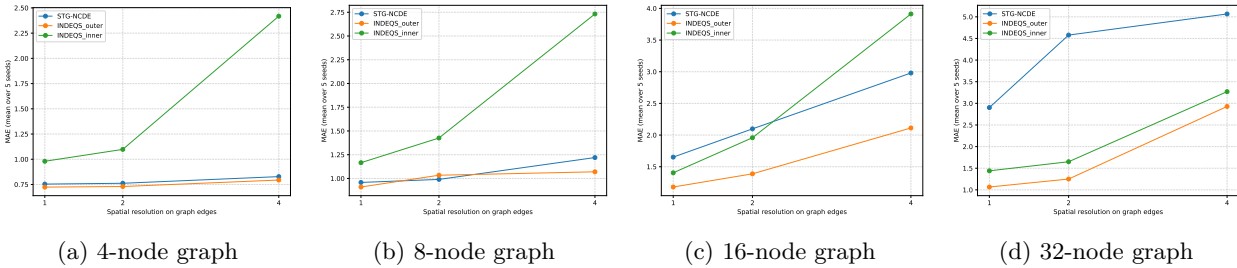

(a) 4-node graph        (b) 8-node graph        (c) 16-node graph        (d) 32-node graph

Figure 7: Mean MAE over 5 runs for increasing spatial node resolution for the **Advection experiment** with fixed graph size.

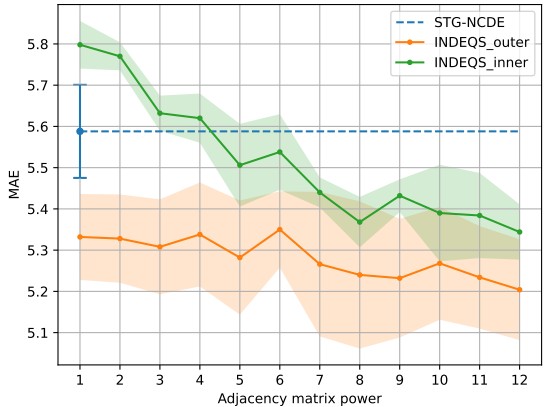

Figure 8: Average performance of INDEQS with increasing adjacency matrix powers vs. the uninformed STG-NCDE for the task of **water discharges** over 5 runs, showing a trend for increased performance for higher adjacencies.

We consider for this task higher adjacency matrix powers and different decoders for the INDEQS methods. The results versus other methods on the water discharge task can be found in Table 1. INDEQS$_{\text{outer}}$ has an MAE of 5.34 compared to uninformed STG-NCDE with MAE of 5.59. We further investigate the forecasting performance for matrix adjacency powers from $k = 1$ up to $k = 12$, the maximum steps one can take upstream, in the water discharge graph. Higher adjacency matrix powers lead to lower MAE for INDEQS$_{\text{inner}}$ and to a smaller degree also for INDEQS$_{\text{outer}}$ (see Figure 8), attaining MAEs of 5.34 and 5.20, respectively, for an adjacency matrix power of 12. INDEQS$_{\text{outer}}$ incurs only marginal overhead in VRAM and compute time, while INDEQS$_{\text{inner}}$ slightly reduces VRAM use and trains 24% faster than STG-NCDE , owing to the lower parameter count of the Adaptive Graph Convolution (AGC). The continuous NODE$_g$ and NODE$_{ctrl}$ decoders lead to additional MAE performance gains for an informed INDEQS$_{\text{outer}}$, improving the MAE of the convolutional decoder of 5.20 to 5.07 and 5.01, respectively — surpassing the uninformed STG-NCDE baseline and the additionally considered informed NODE baseline GCDE-GRU. In Table 1 we further compare to discrete-time graph-informed methods and observe that INDEQS$_{\text{outer}}$ achieves a lower MAE compared to DCRNN but does not match GWaveNet, which achieves the best MAE on the task of water discharges.

Additionally we provide an comparison of graph encoders in relation to different prediction decoders in Section A.9.4. Given the superior performance of INDEQS$_{\text{outer}}$ compared to INDEQS$_{\text{inner}}$, due to the additional data-adaptivity, we focus on the former for the following traffic forecasting task.

Table 1: Comparison of INDEQS vs. the uninformed STG-NCDE and other baselines regarding quantitative performance in terms of averaged MAE and RMSE over five runs in the task of **water discharges** prediction.

| Water discharges | Performance and model specs. | | | | | Model type |
|---|---|---|---|---|---|---|
| | MAE ↓ | RMSE ↓ | #Params. | Time (Epoch/Total) | VRAM | |
| DCRNN (Li et al., 2018) | 5.09 ±0.03 | 16.66 ±0.18 | **23,729** | 13.8 s/ 38.4 min | 117 MB | informed discrete-time RNN |
| GWaveNet (Wu et al., 2019) | **4.95** ±0.13 | **14.89** ±0.42 | 302,537 | **3.6** s/ **5.93** min | 1,808 MB | informed discrete-time CNN |
| STGODE (Fang et al., 2021) | 6.78 ±0.28 | 17.66 ±0.34 | 474,297 | 20.9 s  69.5 min | 204 MB | informed NODE |
| GCDE-GRU (Poli et al., 2021) | 6.28 ±0.29 | 17.75 ±0.37 | 33,281 | 91.7 s/ 62 min | **74** MB | informed NODE |
| STG-NCDE (Choi et al., 2022) | 5.59 ±0.13 | 18.73 ±0.65 | 415,879 | 11.3 s/ 37.8 min | 560 MB | uninformed NCDE |
| INDEQS$_{NODE_{ctrl}}$ (**ours**) | 5.01 ±0.16 | 16.10 ±0.65 | 416,500 | 28.0 s / 93.2 min | 902 MB | informed NCDE |
| INDEQS$_{NODE_g}$ (**ours**) | 5.07 ±0.08 | 16.43 ±0.34 | 415,619 | 17.13 s / 57.1 min | 833 MB | informed NCDE |
| INDEQS$_{inner}(k = 12)$ (**ours**) | 5.34 ±0.07 | 17.73 ±0.64 | 291,909 | 8.5 s / 28.6 min | 535 MB | informed NCDE |
| INDEQS$_{outer}(k = 12)$ (**ours**) | 5.20 ±0.12 | 16.97 ±0.40 | 415,879 | 12.1 s / 40.2 min | 573 MB | informed NCDE |

### 6.2.3 Traffic forecasting

For the PeMS08 traffic flow dataset, we compare INDEQS$_{outer}$ to STG-NCDE to study if the graph information also works for a task that is more influenced by human behavioral patterns and individual decisions, with a less clear causal graph structure. Additionally, the dataset is an established benchmark dataset in spatio-temporal graph forecasting.

Table 2 that While the results show a slight positive trend regarding performance inTable 2 in favor of INDEQS$_{outer}$ on the PeMS08 traffic flow dataset compared to the uninformed STG-NCDE model with a Mean Absolute Error (MAE) over 5 runs of 16.50 vs. 16.55 (see Table 2) and even 16.39 for the best graph adjacency power (see Table 3), statistical testing across five seeds did not indicate a significant difference (see Section A.8 for statistical tests). The results are not directly comparable to the traffic flow forecasting literature, as we report the results over multiple seeds, while in the literature mostly only one seed result is reported.

Table 2: Overview of MAE, number of parameters, training time and the epoch when validation loss last improved, with standard deviations, for uninformed STG-NCDE vs. informed INDEQS$_{outer}$ for the **PeMS08 traffic** prediction over 5 runs with different random seeds.

| | Outer | Inner | MAE (test) | #Params. | MAE (val.) | Train time | Last improvement |
|---|---|---|---|---|---|---|---|
| STG-NCDE | Identity | AGC | 16.55 ±0.20 | 45,152 | 17.00 ±0.22 | 129.31 min ±1.04 | 183.00 ±3.94 |
| INDEQS$_{outer}$ | Informed | AGC | **16.50** ±0.19 | 45,152 | 16.88 ±0.28 | 143.18 min ±1.45 | 169.00 ±11.20 |

We find empirical evidence that INDEQS$_{outer}$ is preferred, when best performance is required and one wants to inform the model with a pre-defined graph structure in addition to having the adaptive graph convolutional component that is able to learn patterns apart from the pre-defined graph. Additional communication between non-connected nodes in the pre-defined graph is possible in this variant. The pre-defined graph structure is in this sense *informing*. In INDEQS$_{inner}$ on the other hand, the pre-defined graph structure is *constraining* the direct communication between nodes according to the pre-defined graph. This can be desired, if one is certain about the underlying process of the data and wants to hinder communication

Table 3: MAE with standard deviation over 5 runs for different outer-inner graph configurations for increasing powers of the adjacency matrix for the **PeMS08 traffic** prediction.

| | Adjacency matrix power | | | | |
|---|---|---|---|---|---|
| | 1 | 2 | 3 | 4 | 5 |
| INDEQS$_{outer}$ | 16.50 ±0.17 | **16.39** ±0.18 | 16.53 ±0.13 | 16.53 ±0.19 | 16.53 ±0.12 |
| STG-NCDE | 16.55 ±0.18 | - | - | - | - |

between unconnected nodes. We provide further experiments regarding continuous decoders, and an analysis of solvers and solver step sizes in Section A.9.6 and Section A.9.7, respectively.

## 7   Discussion

Our empirical study provides several insights into when and how graph-informed NCDE models are beneficial for spatio-temporal forecasting. Across the controlled advection simulations and the river discharge task, we consistently observe that incorporating prior knowledge about the underlying graph structure into the NCDE improves predictive performance over the uninformed baseline STG-NCDE, particularly when the graph encodes physically meaningful flow or connectivity patterns. INDEQS$_{outer}$ with convolutional decoder head is the overall best choice of the different INDEQS variants, if useful graph information is available. INDEQS$_{inner}$ is favorable if one wants to restrict the connectivity between graph nodes to the given graph adjacency matrix, leading to a lighter and faster model with only slightly worse performance than INDEQS$_{outer}$, due to not possessing the data adaptive capabilities of the AGC. For real world problems it is advantageous to use the continuous decoder setups, due to their higher expressivity. However they require more training than the convolutional decoder.

On the advection simulation data, INDEQS$_{outer}$ and INDEQS$_{inner}$ both benefit from access to the true adjacency matrix of the underlying dynamic. Informing the model at the outer position (INDEQS$_{outer}$) leads to robust gains and these increase up to a point, as the number of graph node increases. We suspect gains of the same magnitude for larger graphs. The inner variant (INDEQS$_{inner}$) exploits graph structure more aggressively through a constrained spatial vector field. The inner variant is performing worse than STG-NCDE for higher spatial resolution, but increasingly better for higher numbers of graph nodes. This suggests that for smaller graphs the AGC is compensating for the lacking graph structure and can learn it from data. As the graph becomes larger, the higher the importance of the underlying structure, which cannot be adequately be learned from the data as for smaller graphs. The experiments on varying node resolution indicate that the proposed mechanisms remain effective to a certain degree when the graph is refined. The degrading performance in this case shows that the problem is becoming more difficult, as information now has to travel via multiple hops. The uninformed NCDE, however, is unable to learn the graph structure from the limited given amount of samples, even with clean signal data in the controlled advection simulation setting.

For the river discharge task, INDEQS$_{inner}$ achieves lower MAE than an uninformed NCDE with learnable graph embedding when the adjacency power is chosen appropriately. This highlights that constraining the propagation of information to physically plausible connections can be more beneficial than allowing arbitrary spatial long-range communication, especially in settings where mass conservation and directed flow are central. At the same time, INDEQS$_{outer}$ achieves competitive or better performance while adding only marginal overhead in terms of VRAM and compute time.

On the PeMS08 traffic forecasting benchmark, where the dynamics are influenced by additional unobserved factors and noise, the performance gap between informed and uninformed models is negligible. INDEQS$_{outer}$ performs on par to the uninformed STG-NCDE, indicating that real-world road network information is less useful even when the graph structure does not fully explain the observations.

A second set of insights concerns the choice of decoder. In both the river discharge and traffic experiments, continuous-time decoders (NODE$_g$ and NODE$_{ctrl}$) consistently outperform the discrete convolutional decoder in terms of validation and test MAE. This supports the view that continuous latent dynamics are a natural fit for NCDE-based models on graph time series, especially when predictions at arbitrary time points are of interest. However, these gains come at increased computational cost, most noticebly for NODE$_{ctrl}$, which requires solving an additional ODE for the learned control signal. In practice, NODE$_g$ offers a favorable trade-off between accuracy, parameter count, and runtime. Surprisingly, the continuous decoders do not perform well on the advection simulation.

One obvious limitations is the requirement of knowing the underlying graph structure, which is not always obtainable. Also our evaluations only encompass limited graph types, graph sizes and domains. It is also

difficult to distinguish true underlying graph structure of the dynamic from a mere spatial proximity graph structure, or to discern if a given graph structure is valuable.

Overall, our results indicate that graph-informed NCDEs are particularly beneficial when (i) a reliable graph describing flow or interaction structure is available, (ii) the forecasting horizon and spatial resolution require information to propagate over multiple hops, and (iii) continuous-time evolution of latent trajectories is advantageous. In such settings, INDEQS$_{\text{outer}}$ is a strong choice when performance and flexibility are prioritized, whereas INDEQS$_{\text{inner}}$ is appealing when one wishes to enforce strict adherence to a known topology. We believe that these findings encourage further exploration of informed neural differential equation models for spatio-temporal problems beyond the domains considered in this work.

## 8 Conclusion

We introduced INDEQS, a graph-informed Neural Controlled Differential Equation model that separates inner mixing of hidden states from outer mixing between vector field and control. By injecting prior knowledge of directed graph structure at these distinct positions, INDEQS can achieve lower forecasting errors than uninformed NCDE baselines with comparable parameter counts, particularly on larger graphs and tasks that require information to propagate over multiple hops. Through the introduced controlled advection simulations and the real-world experiment on river discharge, we showed that outer informedness is a strong and flexible default, while inner informedness offers a more parameter-efficient alternative when strict adherence to a known adjacency is desired. Our results highlight the promise of combining neural differential equations with explicit spatial structure, and motivate further work on informed continuous-time models for further spatio-temporal applications.

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

# A   Appendix

## A.1 List of Acronyms

## Acronyms

| | |
|---|---|
| **AGC** | Adaptive Graph Convolution |
| **AST-GCN** | Attribute-Augmented Spatiotemporal Graph Convolutional Network |
| **ASTGCN** | Attention Based Spatial-Temporal Graph Convolutional Networks |
| **CDE** | Controlled Differential Equation |
| **DCRNN** | Diffusion Convolutional Recurrent Neural Network |
| **GCDE-GRU** | Graph Convolutional ODE with Gated Recurrent Units |
| **GCN** | Graph Convolutional Networks |
| **GMAN** | Graph Multi-Attention Network |
| **GN-CDE** | Graph Neural Controlled Differential Equations |
| **GNCDE** | Graph Neural Controlled Differential Equation |
| **GNN** | Graph Neural Network |
| **GNR** | Growing Network with Redirection |
| **GRU** | Gated Recurrent Unit |
| **GWaveNet** | Graph WaveNet |
| **LSTM** | Long Short Term Memory |
| **MAE** | Mean Absolute Error |
| **NCDE** | Neural Controlled Differential Equation |
| **NDE** | Neural Differential Equation |
| **NN** | Neural Network |
| **NODE** | Neural Ordinary Differential Equation |
| **ODE** | Ordinary Differential Equation |
| **PDE** | Partial Differential Equation |
| **ResNet** | Residual Neural Network |
| **RNN** | Recurrent Neural Network |
| **STG-NCDE** | Spatial Temporal Graph Neura Controlled Differential Equations |
| **STGCN** | Spatial-Temporal Graph Convolutional Network |
| **TCN** | Temporal Convolutional Network |

## A.2 Dimensions

The dimensions of the domains and codomains of the vector field functions $f_{\boldsymbol{\theta}}$ and $g_{\boldsymbol{\gamma}}$ can be described even more precisely with

$$f_{\boldsymbol{\theta}}(\cdot) : \mathbb{R}^{|\mathcal{V}| \times d_h} \to \mathbb{R}^{|\mathcal{V}| \times d_h \times d_x}$$

$$g_{\boldsymbol{\gamma}}(\cdot) : \mathbb{R}^{|\mathcal{V}| \times d_z} \to \mathbb{R}^{|\mathcal{V}| \times d_z \times d_h}$$

and $\mathbf{H}(t) \in \mathbb{R}^{|\mathcal{V}| \times d_h}$, $\mathbf{Z}(t) \in \mathbb{R}^{|\mathcal{V}| \times d_z}$, $\mathbf{X}(t) \in \mathbb{R}^{|\mathcal{V}| \times d_x}$, with $d_h$, $d_z$ and $d_x$ being the dimensionality of the corresponding (hidden) states or control per vertex, leading to a description of the tensor product in Equation (1) with einstein notation while omitting the indices of $\boldsymbol{f}$ and $\boldsymbol{g}$ — $\boldsymbol{\theta}$ and $\boldsymbol{\gamma}$ — for more clarity

$$\left(\frac{d\mathbf{H}}{dt}\right)_{mh} = \left(\boldsymbol{f}\frac{d\mathbf{X}}{dt}\right)_{mh} = \boldsymbol{f}_{mhx}\left(\frac{d\mathbf{X}}{dt}\right)_{mx}$$

and for Equation (2)

$$\left(\frac{d\mathbf{Z}}{dt}\right)_{kz} = \left(\boldsymbol{g}\frac{d\mathbf{H}}{dt}\right)_{kz} = \boldsymbol{g}_{kzh}\left(\frac{d\mathbf{H}}{dt}\right)_{kh}, \tag{13}$$

where $n, m$, and $k$ are indices for the vertices, $h$ and $z$ are the indices for the dimension of hidden states $\mathbf{H}$ and $\mathbf{Z}$, respectively and $x$ is the index for the dimension of the control path $\mathbf{X}$.

If one now wants to incorporate graph information at the outer position in Equation (6), one would insert $\mathbf{A}_{\mathcal{V}}$ into Equation (13) and obtain

$$\left(\frac{d\mathbf{Z}}{dt}\right)_{kz} = \left(\boldsymbol{g}\frac{d\mathbf{H}}{dt}\right)_{kz} = \boldsymbol{g}_{kzh}\ (\mathbf{A}_{\mathcal{V}})_{kn}\ \left(\frac{d\mathbf{H}}{dt}\right)_{nh}.$$

### A.3 System of Differential Equations

Starting from the coupled differential equations

$$\begin{bmatrix} \boldsymbol{Z}(T) \\ \boldsymbol{H}(T) \end{bmatrix} = \begin{bmatrix} \boldsymbol{Z}(0) + \int_0^T \boldsymbol{g}_{\gamma}\Big(\boldsymbol{Z}(t)\Big)\frac{d\boldsymbol{H}(t)}{dt} \\ \boldsymbol{H}(0) + \int_0^T \boldsymbol{f}_{\boldsymbol{\theta}}\Big(\boldsymbol{H}(t)\Big)\frac{d\boldsymbol{X}(t)}{dt} \end{bmatrix} \tag{14}$$

with

$$\begin{aligned} \frac{d\boldsymbol{H}(t)}{dt} &= \frac{d}{dt}\left(\mathbf{H}(0) + \int_0^t \boldsymbol{f}_{\theta}\Big(\mathbf{H}(s)\Big)\frac{d\mathbf{X}(s)}{ds}ds\right) \\ &= \frac{d}{dt}\left(\int \boldsymbol{f}_{\theta}\Big(\mathbf{H}(s)\Big)\frac{d\mathbf{X}(s)}{ds}ds\bigg|_{s=t} - \int \boldsymbol{f}_{\theta}\Big(\mathbf{H}(s)\Big)\frac{d\mathbf{X}(s)}{ds}ds\bigg|_{s=0}\right) \\ &= \boldsymbol{f}_{\theta}\left(\mathbf{H}(t)\right)\frac{d\mathbf{X}(t)}{dt}, \end{aligned} \tag{15}$$

it follows that

$$\begin{bmatrix} \boldsymbol{Z}(t) \\ \boldsymbol{H}(t) \end{bmatrix} = \begin{bmatrix} \boldsymbol{Z}(0) + \int_0^T \boldsymbol{g}_{\gamma}\Big(\boldsymbol{Z}(t)\Big)\boldsymbol{f}_{\boldsymbol{\theta}}\Big(\boldsymbol{H}(t)\Big)\frac{d\boldsymbol{X}(t)}{dt} \\ \boldsymbol{H}(0) + \int_0^T \boldsymbol{f}_{\boldsymbol{\theta}}\Big(\boldsymbol{H}(t)\Big)\frac{d\boldsymbol{X}(t)}{dt} \end{bmatrix}. \tag{16}$$

Differentiating and using Equation (15) for $\boldsymbol{H}$ and analogously for $\boldsymbol{Z}$ gives

$$\frac{d}{dt}\begin{bmatrix} \boldsymbol{Z}(t) \\ \boldsymbol{H}(t) \end{bmatrix} = \begin{bmatrix} \boldsymbol{g}_{\gamma}\Big(\boldsymbol{Z}(t)\Big)\boldsymbol{f}_{\boldsymbol{\theta}}\Big(\boldsymbol{H}(t)\Big)\frac{d\boldsymbol{X}(t)}{dt} \\ \boldsymbol{f}_{\boldsymbol{\theta}}\Big(\boldsymbol{H}(t)\Big)\frac{d\boldsymbol{X}(t)}{dt} \end{bmatrix}. \tag{17}$$

### A.4 Edge transition matrix

For the advection simulation in Section 5, we require the edge transition matrix $\mathbf{A}_{\mathcal{E}} \in \mathbb{R}^{|\mathcal{E}|\times|\mathcal{E}|}$, according to which the masses on the edges are distributed to other edges. The underlying graph structure of the dynamic is given by a weighted vertex adjacency matrix $\mathbf{A}_{\mathcal{V}}$ with $\sum_{j\in|\mathcal{V}|}(\mathbf{A}_{\mathcal{V}})_{ij} = 1$. One can always convert a directed vertex adjacency matrix without self-loops to a signed vertex incidence matrix $\mathbf{I}$. Let us define some matrix operations for $\mathbf{I} \in \mathbb{R}^{|\mathcal{V}|\times|\mathcal{E}|}$, namely

$$(\mathbf{I}^+)_{ij} = \begin{cases} (\mathbf{I})_{ij}, & \text{if } (\mathbf{I})_{ij} > 0 \\ 0, & \text{else} \end{cases},$$

$$(\mathbf{I}^-)_{ij} = \begin{cases} -(\mathbf{I})_{ij}, & \text{if } (\mathbf{I})_{ij} < 0 \\ 0, & \text{else} \end{cases},$$

$$(\mathbf{I}^c)_{ij} = \begin{cases} 1, & \text{if } (\mathbf{I})_{ij} > 0 \\ (\mathbf{I})_{ij}, & \text{else} \end{cases},$$

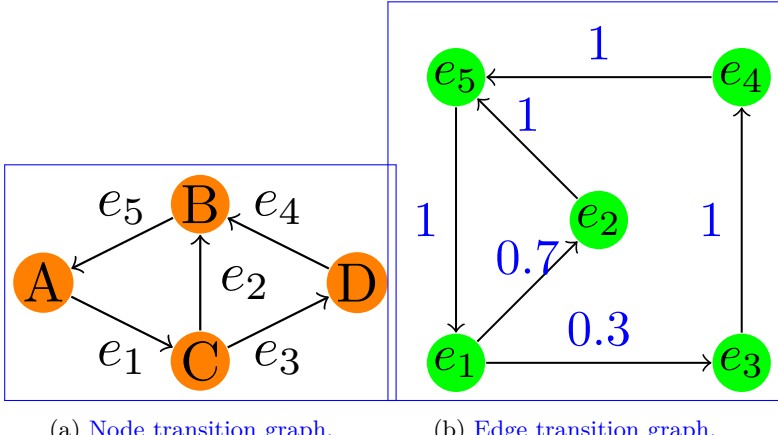

(a) Node transition graph.    (b) Edge transition graph.

Figure 9: Given a node transition graph, one can obtain the corresponding edge transition graph.

thereby $(\cdot)^c$ is making $\mathbf{I}$ conservative. In total the edge transition matrix $\mathbf{A}_{\mathcal{E}}$ is given by

$$\mathbf{A}_{\mathcal{E}} = (\mathbf{I}^-)^\intercal (\mathbf{I}^c)^+.$$

On the one side $(\mathbf{I}^-)^\intercal \in \mathbb{R}^{|\mathcal{E}| \times |\mathcal{V}|}$ considers the incoming magnitude of the transported quantities from edges to vertices and also the amount of quantities, on the other side $(\mathbf{I}^c)^+ \in \mathbb{R}^{|\mathcal{V}| \times |\mathcal{E}|}$ distributes the outgoing quantites from the vertices to the edges while solely encoding connectivity and neglecting magnitude.

For clarity, in the following we provide all the matrices for the conversion of an exemplary vertex adjacency graph to an edge transition graph, both depicted in Figure 9, which is the underlying structure of the dynamic in Figure 4:

$$\mathbf{A}_{\mathcal{V}} = \begin{pmatrix} 0 & 0 & 1 & 0 \\ 1 & 0 & 0 & 0 \\ 0 & 0.7 & 0 & 0.3 \\ 0 & 1 & 0 & 0 \end{pmatrix}, \qquad \mathbf{I} = \begin{pmatrix} -1 & 0 & 0 & 0 & 1 \\ 1 & 0.7 & 0 & 1 & -1 \\ 1 & -0.7 & -0.3 & 0 & 0 \\ 0 & 0 & 0.3 & -1 & 0 \end{pmatrix}$$

$$\mathbf{I}^- = \begin{pmatrix} 1 & 0 & 0 & 0 & 0 \\ 0 & 0 & 0 & 0 & 1 \\ 0 & 0.7 & 0.3 & 0 & 0 \\ 0 & 0 & 0 & 1 & 0 \end{pmatrix}, \quad \mathbf{I}^c = \begin{pmatrix} -1 & 0 & 0 & 0 & 1 \\ 0 & 1 & 0 & 1 & -1 \\ 1 & -0.7 & -0.3 & 0 & 0 \\ 0 & 0 & 1 & -1 & 0 \end{pmatrix}, \quad (\mathbf{I}^c)^+ = \begin{pmatrix} 0 & 0 & 0 & 0 & 1 \\ 0 & 1 & 0 & 1 & 0 \\ 1 & 0 & 0 & 0 & 0 \\ 0 & 0 & 1 & 0 & 0 \end{pmatrix}.$$

This leads to the edge transition matrix

$$\mathbf{A}_{\mathcal{E}} = (\mathbf{I}^-)^\intercal (\mathbf{I}^c)^+ = \begin{pmatrix} 0 & 0 & 0 & 0 & 1 \\ 0.7 & 0 & 0 & 0 & 0 \\ 0.3 & 0 & 0 & 0 & 0 \\ 0 & 0 & 1 & 0 & 0 \\ 0 & 1 & 0 & 1 & 0 \end{pmatrix}.$$

### A.5 Generating Growing Network with Redirection Graph

The GNR graph is built with networkx (Hagberg et al., 2008) by sequentially adding nodes with a link to one previously added node. The previous target node is chosen uniformly at random. With probability $p$ the edge connection is instead "redirected" to the successor node of the target. A probability $p$ closer to 1 leads to broader networks, rather having many edges leading in one central node, while $p$ closer to 0 leads to deeper networks with longer chains leading into a central node. We use an redirection probability of $p = 0$ to attain a deeper network that resembles a river network more closely.

## A.6   Causality

In total we moved from a space-continuous description of the state on the edges and vertices during the simulation to a discrete space on the vertices for the measurements. In the continuous advection description, we had a clear cause-effect relation due to the fact that we could look back along an edge or back to previous edges with the edge transition matrix. This continuous description of the causal structure is now subsumed to aggregated, discrete values at nodes and edge connections without an attached domain between them. To be able to incorporate this causal structure into the GNCDEs we have to consider a long enough context window to be able to receive information from the previous vertices at earlier times. The task then resembles learning a delay differential equation.

## A.7   Continuous decoder architectures

For forecasting future time steps of the target feature we propose two alternative decoder architectures based on continuous-time extrapolation using NODEs. Both decoder variants replace the final convolutional decoder (see Figure 10), but differ in how the continuous extrapolation is learned: either directly in latent space as a continuation of $\boldsymbol{Z}(t)$ or through a learned extrapolation of the control path $\boldsymbol{X}(t)$ in real space. The decoder variants allow generating predictions within an NCDE framework, where the control path is unknown over the forecasting horizon.

### A.7.1   NODE$_g$ continuous decoder.

The first decoder we propose uses $\boldsymbol{Z}(T)$ as the initial condition to an additional NODE whose solution is evaluated at future times $T < t_i \leq T + \tau$ to obtain the final predictions $\{\hat{\mathbf{Y}}_{t_i}\}_{i=1}^{M}$ for the next $M$ time steps up to time $T + \tau$ (see Section A.7.2). This yields the continuous-time evolution

$$\boldsymbol{P}(t) = \boldsymbol{Z}(T) + \int_{T}^{T+t} \boldsymbol{g}_{\gamma}\Big(\boldsymbol{P}(t')\Big)\mathbf{1}\, dt', \tag{18}$$

where $\mathbf{1} \in \mathbb{R}^{|\mathcal{V}| \times d_h}$ is the matrix of ones and the vector field $\boldsymbol{g}_{\gamma}$ shares its parameters with the encoding $\boldsymbol{g}_{\gamma}$ in Equation (3). The product in the integrand $\boldsymbol{g}_{\gamma}\mathbf{1}$ is a batched matrix-vector multiplication over the node dimension $(\boldsymbol{g}_{\theta}\mathbf{1})_{mz} = (\boldsymbol{g}_{\theta})_{mzh}(\mathbf{1})_{mh}$. Through $\boldsymbol{g}_{\gamma}$, the continued trajectory $\boldsymbol{P}(t)$ contains graph information, depending on the informedness of the inner vector field $\boldsymbol{g}_{\gamma}$. Predictions are obtained by evaluating $\boldsymbol{P}(t_i)$ in latent space at times $T < t_i \leq T + \tau$ and projecting to the data space via a learnable affine layer $l_{\delta} : \mathbb{R}^{|\mathcal{V}| \times d_z} \to \mathbb{R}^{|\mathcal{V}| \times d_x}$, $\boldsymbol{P}(t_i) \mapsto l_{\delta}(\boldsymbol{P}(t_i)) = \hat{\mathbf{Y}}_{t_i}$. The choice of $\mathbf{1}$ in Equation (18) reflects the absence of an external control signal beyond time point $T$. While this is the simplest option, alternative formulations could instead incorporate a learnable control signal. An alternative approach, including such a learnable control is introduced in the following.

### A.7.2   NODE$_{ctrl}$ continuous decoder.

For the second decoder type, instead of evolving from $\boldsymbol{Z}(T)$ by a separate NODE and thereby temporally extending the latent space, the observed control $\boldsymbol{X}(t)$ is extended beyond time $T$ by an NODE which learns the vector field $\boldsymbol{h}_{\boldsymbol{\lambda}}$ from data up till $T$. The state $\boldsymbol{C}(t)$ can be described by

$$\boldsymbol{C}(t) = \boldsymbol{X}(0) + \int_{0}^{t} \boldsymbol{h}_{\boldsymbol{\lambda}}(t', \boldsymbol{C}(t'))\, dt'. \tag{19}$$

To drive the total system, one uses the piecewise control

$$\bar{\boldsymbol{X}}(t) = \begin{cases} \boldsymbol{X}(t), & 0 \leq t \leq T, \\ \boldsymbol{C}(t), & T < t \leq T + \tau. \end{cases} \tag{20}$$

Using $\bar{\boldsymbol{X}}(t)$ in place of the original control yields the jointly coupled system

$$\frac{d}{dt} \begin{bmatrix} \boldsymbol{Z}(t) \\ \boldsymbol{H}(t) \\ \boldsymbol{C}(t) \end{bmatrix} = \begin{bmatrix} \boldsymbol{g}_{\gamma}\Big(\boldsymbol{Z}(t)\Big)\boldsymbol{f_\theta}\Big(\boldsymbol{H}(t)\Big)\frac{d\bar{\boldsymbol{X}}(t)}{dt} \\ \boldsymbol{f_\theta}\Big(\boldsymbol{H}(t)\Big)\frac{d\bar{\boldsymbol{X}}(t)}{dt} \\ \boldsymbol{h_\lambda}\Big(t, \boldsymbol{C}(t)\Big) \end{bmatrix}. \tag{21}$$

Accordingly, the hidden states $\boldsymbol{H}(t)$ and $\boldsymbol{Z}(t)$ are driven by the observed path on $[0, T]$ and by the learned continuation on $(T, T+\tau]$. Forecasts are again obtained by evaluating the evolved latent states $\boldsymbol{Z}(t_i)$ at times $T < t_i \le T + \tau$ and mapping them to the target space via a learnable linear layer $\boldsymbol{Z}(t_i) \mapsto l_\delta(\boldsymbol{Z}(t_i)) = \hat{\boldsymbol{Y}}_{t_i}$, analogous to the $\text{NODE}_g$ decoder (see Section A.7.1). Since, in this decoder setting, the NODE of $\boldsymbol{C}(t)$ is trained in parallel to the NCDE, we employ two loss terms: the standard NCDE loss and an additional loss $\mathcal{L}(\{\mathbf{C}_{t_j}\}, \{\mathbf{X}_{t_j}\})$ on the control, which is evaluated over the entire time window $0 \le t_j \le T + \tau$. During training, this term is scaled by a factor of 1/epoch, such that the control loss receives a substantially higher relative weight at the beginning of training, which gradually decreases towards the end.

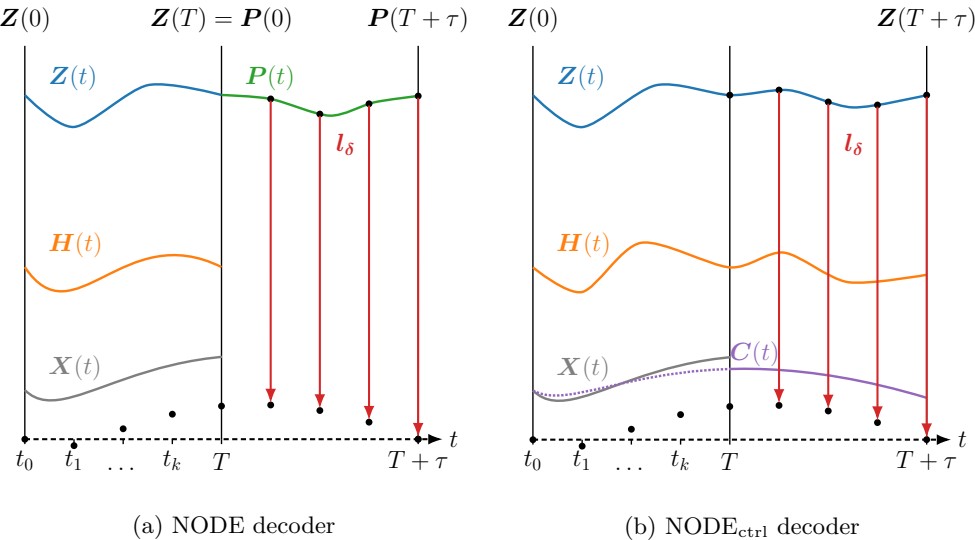

(a) NODE decoder  (b) $\text{NODE}_{\text{ctrl}}$ decoder

Figure 10: Graphical overview of the NCDE encoder-decoder structure in combination with two continuous decoder variants on the right halves in (a) and (b). **(a)**: $\text{NODE}_g$-decoder evolves a hidden state $\boldsymbol{P}$ for $t > T$ via an NODE, based on the last hidden state $\boldsymbol{Z}(T)$ over time via an NODE. Predictions can be read out at any given time via an affine layer (red). **(b)**: $\text{NODE}_{ctrl}$-decoder allows for continuous evolution of $\boldsymbol{Z}$ and $\boldsymbol{H}$ into the prediction time window by learning an extrapolation $\boldsymbol{C}$ of the control signal $\boldsymbol{X}$. The readout of predictions is identical to $\text{NODE}_g$.

## A.8  Statistical tests

For the traffic flow prediction results on PEMS08, we perform a two-sided Wilcoxon signed-rank test to verify the significance of our proposed methods against the baseline across 5 independent seeds. Specifically, the comparison between STG-NCDE and $\text{INDEQS}_{\text{outer}}$ yields $p = 0.563$, while the comparison with $\text{INDEQS}_{\text{outer}}(k=2)$ yields $p = 0.625$. This indicates that the observed performance gains cannot be confidently distinguished from random variation.

## A.9  Additional Experimental Results

In this section we provide further material regarding the experimental results for the advection data, river discharge task and the PeMS08 traffic flow dataset, and results regarding the different Decoder settings.

## A.10

### A.9.1  Advection simulation data

For the advection simulation data, we additionally provide the table results of the variants of the main part in Table 4. Moreover we compared further outer-inner graph informedness variations whose results can be found in Table 5.

Table 4: MAE for different outer-inner informedness mechanisms for graphs with increasing graph nodes, and increasing node resolution along the edges

| | Outer | Inner | Nodes | Spatial resolution | | |
|---|---|---|---|---|---|---|
| | | | | 1 | 2 | 4 |
| STG-NCDE | Identity | AGC | 4 | $0.75$ ±0.0513 | $0.76$ ±0.0526 | $0.83$ ±0.0926 |
| STG-NCDE | Identity | AGC | 8 | $0.96$ ±0.0476 | $0.99$ ±0.0678 | $1.22$ ±0.1951 |
| STG-NCDE | Identity | AGC | 16 | $1.65$ ±0.1424 | $2.10$ ±0.2567 | $2.98$ ±0.4047 |
| STG-NCDE | Identity | AGC | 32 | $2.90$ ±0.2319 | $4.58$ ±0.7343 | $5.07$ ±0.4969 |
| STG-NCDE | Identity | AGC | 64 | $5.64$ ±0.1174 | – | – |
| STG-NCDE | Identity | AGC | 128 | $5.42$ ±0.0614 | – | – |
| INDEQS$_{outer}$ | Informed | AGC | 4 | $0.72$ ±0.0297 | $0.73$ ±0.0524 | $0.79$ ±0.0422 |
| INDEQS$_{outer}$ | Informed | AGC | 8 | $0.91$ ±0.0292 | $1.03$ ±0.0680 | $1.07$ ±0.0624 |
| INDEQS$_{outer}$ | Informed | AGC | 16 | $1.18$ ±0.0806 | $1.39$ ±0.0563 | $2.11$ ±0.1038 |
| INDEQS$_{outer}$ | Informed | AGC | 32 | $1.07$ ±0.0981 | $1.25$ ±0.0579 | $2.93$ ±0.3218 |
| INDEQS$_{outer}$ | Informed | AGC | 64 | $1.01$ ±0.0563 | – | – |
| INDEQS$_{outer}$ | Informed | AGC | 128 | $0.89$ ±0.0391 | – | – |
| INDEQS$_{inner}$ | Identity | Informed | 4 | $0.92$ ±0.0716 | $1.07$ ±0.1482 | $2.01$ ±0.1266 |
| INDEQS$_{inner}$ | Identity | Informed | 8 | $1.11$ ±0.0716 | $1.40$ ±0.0868 | $2.50$ ±0.1993 |
| INDEQS$_{inner}$ | Identity | Informed | 16 | $1.41$ ±0.1144 | $1.73$ ±0.2232 | $3.36$ ±0.4732 |
| INDEQS$_{inner}$ | Identity | Informed | 32 | $1.39$ ±0.2397 | $1.55$ ±0.1440 | $3.10$ ±0.5847 |
| INDEQS$_{inner}$ | Identity | Informed | 64 | $1.44$ ±0.2252 | – | – |
| INDEQS$_{inner}$ | Identity | Informed | 128 | $1.60$ ±0.2624 | – | – |

### A.9.2  Robustness analysis to inaccurate graph priors

We further investigate the effect of incorrect graph information on the advection simulation prediction by removing an increasing percentage of non-self-loop edges or adding previously non-existent non-self-loop edges, starting from the correct underlying graph. This makes the graph used for training increasingly different compared to the original correct graph for both directions. Two additional scenarios, in which we transpose the correct graph or add an edge in the opposite direction for every given edge, is also created. Based on these graphs, we train and evaluate the predictive performance. The results of these experiments can be seen in Figure 11 and Table 6. INDEQS$_{outer}$'s MAE is consistently lower than the MAE of STG-NCDE in our graph perturbation experiments until it coincides with STG-NCDE for a reduction of 100% of the edges, as this case signifies informing with an identity matrix at the outer position. INDEQS$_{inner}$ also is possessing lower MAE up to a reduction of about 80% of the edges and then collapses to a an NCDE with both the inner and outer informedness being the identity, leading to no mixing between nodes. Adding incorrect edges seems to be less detrimental as removing correct edges for both INDEQS variations.

### A.9.3  Water discharges

For the water discharge task, we additionally compared different mechanisms at the outer and inner positions shown in Table 7 and different adjacency matrix powers in Figure 12 and Table 8.

### A.9.4  Systematic comparison of graph informedness vs. continuous decoders

To separate the effect of the graph informed encoder from the continuous prediction decoders, we conducted experiments with different encoder-decoder variations on the river discharge data. The results can be found in Table 9.

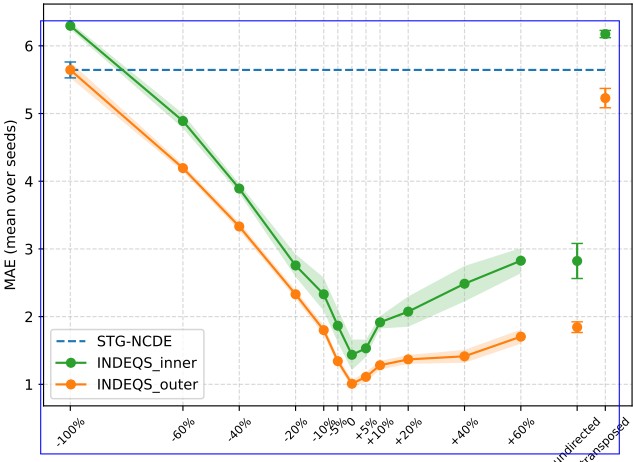

Figure 11: MAE for the graph advection simulation on 64 nodes for INDEQS$_{outer}$ and INDEQS$_{inner}$ for increasingly missing and additional incorrect edges over 5 seeds.

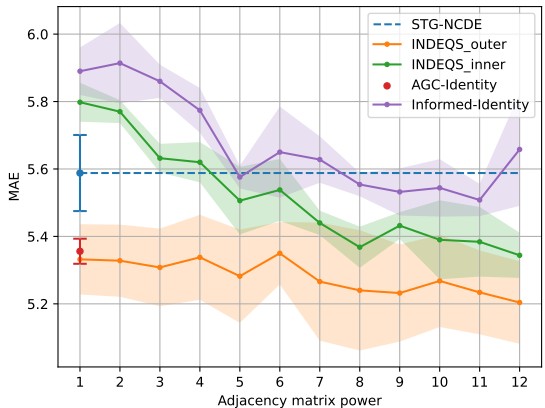

Figure 12: Average performance of INDEQS and additional informedness variations, with increasing adjacency matrix powers vs. the uninformed STG-NCDE for the task of water discharges over 5 runs.

### A.9.5 PeMS08 traffic flow

For the traffic flow task, we show results and properties of 2 additional outer-inner graph information strategies in Table 10 and also results for higher adjacency matrix powers in Table 11.

### A.9.6 Result comparison among decoder types

We compare the discrete convolutional decoder with the continuous decoder architectures introduced in Section A.7, namely NODE$_g$ and NODE$_{ctrl}$, on the advection simulation, river discharge and traffic forecasting tasks. The corresponding results are reported in Tables 12-15. In both real-world settings, the continuous decoders outperform the discrete convolutional decoder in terms of both validation and test MAE, indicating the advantage of continuous-time decoding for these tasks. While NODE$_{ctrl}$ achieves the best performance on the river discharge task, NODE$_g$ appears, on average, to be the strongest option for traffic forecasting.

A practical advantage of NODE$_g$ over NODE$_{ctrl}$ is that it improves MAE without introducing additional hyperparameters, while also requiring fewest trainable parameters. More generally, the superior performance of the continuous decoder architectures comes at the cost of increased computational time, since in both cases the hidden-state trajectories $Z$ must also be solved over the prediction window. This effect is most pronounced for NODE$_{ctrl}$, as the NODE governing the learned control signal $C$ must be solved in addition

to the continuation of the hidden-state trajectories $\boldsymbol{Z}$ and $\boldsymbol{H}$.

For the advection simulation task however, the continuous decoders perform significantly worse than the convolutional prediction head (see Table 12 and Table 13).

### A.9.7   Comparison of solvers and step-sizes for all decoder architectures

We compare the three decoder architectures on the water discharge task across combinations of numerical solvers and solver step sizes. The results are shown in Figure 13, which reports validation MAE and training time as a function of the solver step size. Across all decoders, we observe a solver-specific optimal step size that yields the best trade-off between predictive performance and computational cost. Decreasing the step size beyond this point consistently increases training time, while providing little or no additional improvement in MAE. For the smallest step sizes, *Euler* and *RK4* achieve comparable MAE, although *RK4* remains substantially more expensive computationally. To ensure consistency across experiments, we use *RK4* with step size 1 in all main results. However, if the best performance needs to be achieved, the solver configuration may depend on the decoder architecture. For example, the NODE$_{ctrl}$ decoder (Fig. 13, right) appears to perform best with the computationally cheaper *Euler* solver at step size 0.5, while the convolutional decoder (Fig. 13, left) may benefit from *Euler* with step size 0.25 and NODE$_g$ may be used with either *Euler* 0.25 or *RK4* 1.0.

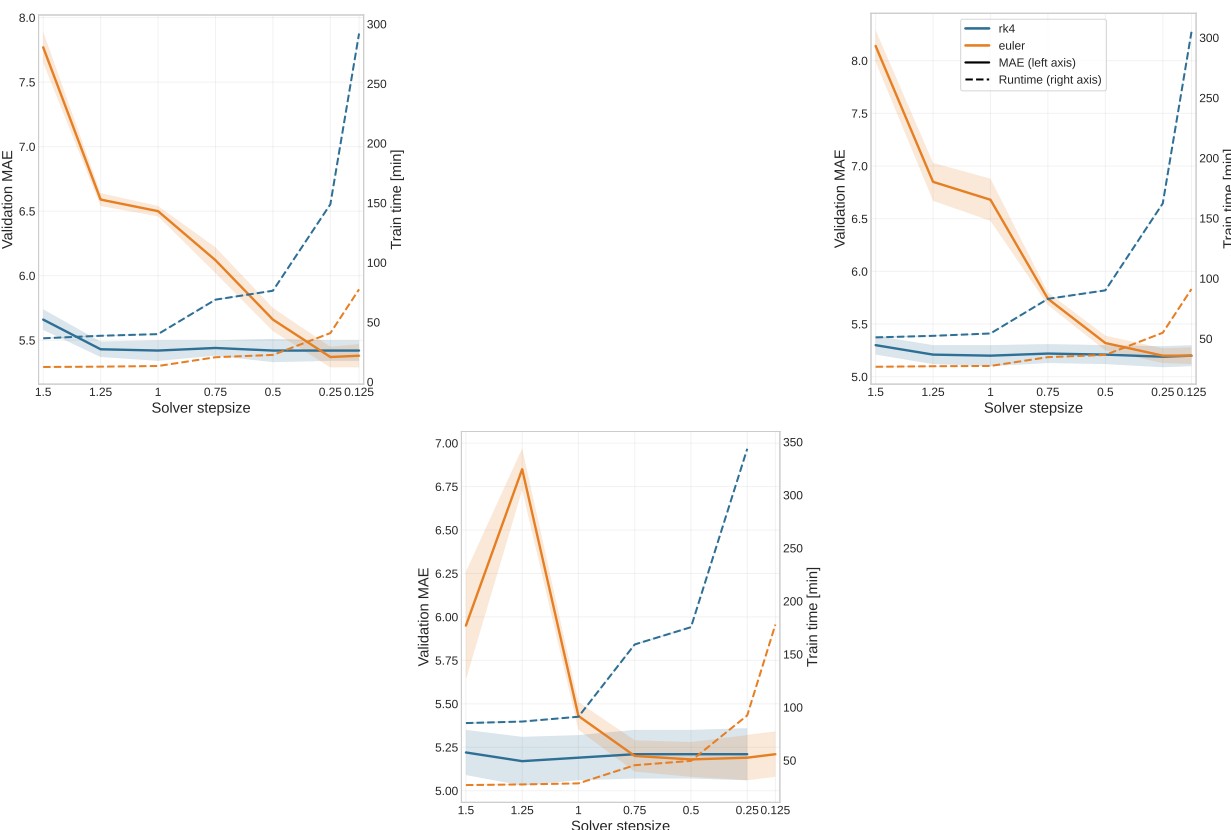

Figure 13: MAE and runtime for different solver and step-size combinations for the discrete convolutional decoder (left), the continuous NODE$_g$ decoder (center) and the continuous NODE$_{ctrl}$ decoder (right). All experiments were performed on the **water discharge** prediction task and run for 200 epochs. Standard deviation over 5 runs is displayed as shaded area.

### A.10 Software

For data preparation *numpy* (Harris et al., 2020), *scipy* (Virtanen et al., 2020) and *pandas* (Wes McKinney, 2010; pandas development team, 2020) was used. Plots were created with *matplotlib* (Hunter, 2007). The model code is based on Choi et al. (2022)'s code and relies on *pytorch* (Paszke et al., 2017; 2019), *torchdiffeq* (Chen, 2018) and *controldiffeq* (Kidger et al., 2020). To monitor training runs we used *tensorboard* (Abadi et al., 2015). The cluster for the experiments ran images of *Ubuntu 20.04* with *NVIDIA CUDA 11.7.1* and *CUDNN 9*.

### A.11 Hardware

All experiments were conducted on a cluster node with an AMD EPYC 73F3 3.5 GHz with 16 cores and 64 GB RAM, and a NVIDIA A100 with 40 GB VRAM.

Table 5: MAE for different outer-inner informedness mechanisms for graphs with increasing graph nodes, and increasing node resolution along the edges

| Label | Outer | Inner | Nodes | Spatial resolution | | |
|---|---|---|---|---|---|---|
| | | | | 1 | 2 | 4 |
| | AGC | AGC | 4 | 0.79 ±0.0321 | 0.72 ±0.0358 | 0.85 ±0.0850 |
| | AGC | AGC | 8 | 0.98 ±0.0614 | 1.05 ±0.1154 | 2.51 ±0.9738 |
| | AGC | AGC | 16 | 1.66 ±0.3254 | 3.70 ±1.1690 | 5.08 ±0.6714 |
| | AGC | AGC | 32 | 4.49 ±1.1091 | 5.33 ±0.0875 | 5.41 ±0.0370 |
| | AGC | AGC | 64 | 5.78 ±0.0332 | – | – |
| | AGC | AGC | 128 | 5.52 ±0.0324 | – | – |
| | AGC | Identity | 4 | 1.41 ±0.3518 | 1.19 ±0.1933 | 1.50 ±0.3887 |
| | AGC | Identity | 8 | 2.23 ±0.7419 | 1.79 ±0.6673 | 1.80 ±0.2517 |
| | AGC | Identity | 16 | 3.08 ±0.5545 | 2.99 ±0.8873 | 4.27 ±1.1252 |
| | AGC | Identity | 32 | 3.33 ±0.4909 | 4.41 ±0.9618 | 5.34 ±0.0546 |
| | AGC | Identity | 64 | 5.24 ±0.5101 | – | – |
| | AGC | Identity | 128 | 5.46 ±0.0434 | – | – |
| | AGC | informed | 4 | 0.80 ±0.0760 | 0.94 ±0.0750 | 0.91 ±0.0909 |
| | AGC | Informed | 8 | 1.01 ±0.0251 | 1.19 ±0.0274 | 1.29 ±0.3112 |
| | AGC | Informed | 16 | 1.38 ±0.0502 | 1.67 ±0.0789 | 2.90 ±0.0532 |
| | AGC | Informed | 32 | 1.56 ±0.0428 | 1.76 ±0.0430 | 2.85 ±0.1536 |
| | AGC | Informed | 64 | 2.22 ±0.0421 | – | – |
| | AGC | Informed | 128 | 2.44 ±0.0316 | – | – |
| STG-NCDE | Identity | AGC | 4 | 0.75 ±0.0513 | 0.76 ±0.0526 | 0.83 ±0.0926 |
| STG-NCDE | Identity | AGC | 8 | 0.96 ±0.0476 | 0.99 ±0.0678 | 1.22 ±0.1951 |
| STG-NCDE | Identity | AGC | 16 | 1.65 ±0.1424 | 2.10 ±0.2567 | 2.98 ±0.4047 |
| STG-NCDE | Identity | AGC | 32 | 2.90 ±0.2319 | 4.58 ±0.7343 | 5.07 ±0.4969 |
| STG-NCDE | Identity | AGC | 64 | 5.64 ±0.1174 | – | – |
| STG-NCDE | Identity | AGC | 128 | 5.42 ±0.0614 | – | – |
| | Identity | Identity | 4 | 4.05 ±0.0451 | 4.03 ±0.0421 | 4.03 ±0.0390 |
| | Identity | Identity | 8 | 4.92 ±0.0167 | 4.92 ±0.0274 | 4.93 ±0.0351 |
| | Identity | Identity | 16 | 6.21 ±0.0365 | 6.21 ±0.0472 | 6.25 ±0.0217 |
| | Identity | Identity | 32 | 5.89 ±0.0179 | 5.92 ±0.0327 | 5.98 ±0.0474 |
| | Identity | Identity | 64 | 6.30 ±0.0513 | – | – |
| | Identity | Identity | 128 | 6.25 ±0.0292 | – | – |
| INDEQS$_{inner}$ | Identity | Informed | 4 | 0.92 ±0.0716 | 1.07 ±0.1482 | 2.01 ±0.1266 |
| INDEQS$_{inner}$ | Identity | Informed | 8 | 1.11 ±0.0716 | 1.40 ±0.0868 | 2.50 ±0.1993 |
| INDEQS$_{inner}$ | Identity | Informed | 16 | 1.41 ±0.1144 | 1.73 ±0.2232 | 3.36 ±0.4732 |
| INDEQS$_{inner}$ | Identity | Informed | 32 | 1.39 ±0.2397 | 1.55 ±0.1440 | 3.10 ±0.5847 |
| INDEQS$_{inner}$ | Identity | Informed | 64 | 1.44 ±0.2252 | – | – |
| INDEQS$_{inner}$ | Identity | Informed | 128 | 1.60 ±0.2624 | – | – |
| INDEQS$_{outer}$ | Informed | AGC | 4 | 0.72 ±0.0297 | 0.73 ±0.0524 | 0.79 ±0.0422 |
| INDEQS$_{outer}$ | Informed | AGC | 8 | 0.91 ±0.0292 | 1.03 ±0.0680 | 1.07 ±0.0624 |
| INDEQS$_{outer}$ | Informed | AGC | 16 | 1.18 ±0.0806 | 1.39 ±0.0563 | 2.11 ±0.1038 |
| INDEQS$_{outer}$ | Informed | AGC | 32 | 1.07 ±0.0981 | 1.25 ±0.0579 | 2.93 ±0.3218 |
| INDEQS$_{outer}$ | Informed | AGC | 64 | 1.01 ±0.0563 | – | – |
| INDEQS$_{outer}$ | Informed | AGC | 128 | 0.89 ±0.0391 | – | – |
| | Informed | Identity | 4 | 0.98 ±0.0822 | 1.10 ±0.1529 | 2.42 ±0.7735 |
| | Informed | Identity | 8 | 1.17 ±0.1452 | 1.43 ±0.3069 | 2.73 ±0.7580 |
| | Informed | Identity | 16 | 1.40 ±0.1421 | 1.96 ±0.2817 | 3.91 ±0.8507 |
| | Informed | Identity | 32 | 1.44 ±0.3059 | 1.65 ±0.1366 | 3.27 ±0.6967 |
| | Informed | Identity | 64 | 1.49 ±0.1657 | – | – |
| | Informed | Identity | 128 | 1.57 ±0.2370 | – | – |
| | Informed | Informed | 4 | 0.91 ±0.1468 | 0.98 ±0.1601 | 1.80 ±0.5672 |
| | Informed | Informed | 8 | 1.02 ±0.0843 | 1.50 ±0.2582 | 2.17 ±0.2936 |
| | Informed | Informed | 16 | 1.29 ±0.1230 | 1.76 ±0.1038 | 2.80 ±0.2686 |
| | Informed | Informed | 32 | 1.50 ±0.1785 | 1.94 ±0.2183 | 2.35 ±0.1228 |
| | Informed | Informed | 64 | 1.59 ±0.2592 | – | – |
| | Informed | Informed | 128 | 1.63 ±0.2187 | – | – |

Table 6: MAE for the graph advection simulation on 64 nodes for INDEQS$_{outer}$ and INDEQS$_{inner}$ for increasingly missing and additional incorrect edges over 5 seeds.

|  | STG-NCDE | INDEQS$_{inner}$ | INDEQS$_{outer}$ |
|---|---|---|---|
| -100% | 5.64 $\pm$0.1174 | – | 5.64 $\pm$0.1174 |
| -60% | – | 4.89 $\pm$0.1079 | 4.19 $\pm$0.0513 |
| -40% | – | 3.89 $\pm$0.0705 | 3.33 $\pm$0.0589 |
| -20% | – | 2.76 $\pm$0.1670 | 2.33 $\pm$0.0612 |
| -10% | – | 2.33 $\pm$0.2449 | 1.80 $\pm$0.0704 |
| -5% | – | 1.87 $\pm$0.2473 | 1.34 $\pm$0.0610 |
| 0 | 5.64 $\pm$0.1174 | 1.44 $\pm$0.2252 | 1.01 $\pm$0.0563 |
| +5% | – | 1.53 $\pm$0.1287 | 1.11 $\pm$0.0689 |
| +10% | – | 1.92 $\pm$0.0913 | 1.28 $\pm$0.0642 |
| +20% | – | 2.07 $\pm$0.2239 | 1.37 $\pm$0.0653 |
| +40% | – | 2.49 $\pm$0.2616 | 1.41 $\pm$0.0953 |
| +60% | – | 2.83 $\pm$0.1861 | 1.70 $\pm$0.1043 |
| undirected | – | 2.82 $\pm$0.2590 | 1.84 $\pm$0.0789 |
| transposed | – | 6.17 $\pm$0.0532 | 5.23 $\pm$0.1427 |

Table 7: Overview of MAE, number of parameters, training time and epoch of last improvement, with standard deviations, for different outer-inner configurations for the **water discharge** prediction over 5 runs with different random seeds.

|  | Outer | Inner | MAE | # Params. | MAE (val.) | Train time | Last improvement |
|---|---|---|---|---|---|---|---|
|  | AGC | Identity | 5.36 $\pm$0.04 | 415,879 | 5.31 $\pm$0.05 | 40.11 min $\pm$0.25 | 191.20 $\pm$3.27 |
|  | AGC | LGC | 5.43 $\pm$0.10 | 539,399 | 5.36 $\pm$0.09 | 50.34 min $\pm$0.41 | 161.20 $\pm$39.95 |
|  | AGC | Informed | **5.33** $\pm$0.04 | 415,879 | 5.31 $\pm$0.05 | 39.97 min $\pm$0.14 | 186.00 $\pm$7.11 |
| STG-NCDE | Identity | AGC | 5.59 $\pm$0.13 | 415,879 | 5.64 $\pm$0.18 | 37.85 min $\pm$0.13 | 188.40 $\pm$10.92 |
|  | Identity | Identity | 5.94 $\pm$0.01 | 291,909 | 6.13 $\pm$0.01 | 28.06 min $\pm$0.14 | 175.60 $\pm$8.08 |
|  | Identity | LGC | 5.86 $\pm$0.03 | 415,429 | 5.94 $\pm$0.02 | 38.34 min $\pm$0.13 | 128.00 $\pm$16.23 |
| INDEQS$_{inner}$ | Identity | Informed | 5.80 $\pm$0.06 | 291,909 | 5.92 $\pm$0.03 | 28.02 min $\pm$0.12 | 177.40 $\pm$26.52 |
| INDEQS$_{outer}$ | Informed | AGC | 5.33 $\pm$0.12 | 415,879 | 5.43 $\pm$0.09 | 40.14 min $\pm$0.09 | 167.60 $\pm$29.42 |
|  | Informed | Identity | 5.89 $\pm$0.08 | 291,909 | 5.99 $\pm$0.06 | 30.28 min $\pm$0.06 | 181.00 $\pm$14.00 |
|  | Informed | LGC | 5.95 $\pm$0.02 | 415,429 | 6.01 $\pm$0.03 | 40.40 min $\pm$0.23 | 88.00 $\pm$10.02 |

Table 8: MAE with standard deviation over 5 runs for different outer-inner graph configurations for increasing powers of the adjacency matrix for the **water discharge** prediction.

|  | Adjacency matrix power | | | | | | | | | | | |
|---|---|---|---|---|---|---|---|---|---|---|---|---|
|  | 1 | 2 | 3 | 4 | 5 | 6 | 7 | 8 | 9 | 10 | 11 | 12 |
| Informed - Identity | 5.89 $\pm$0.07 | 5.91 $\pm$0.12 | 5.86 $\pm$0.05 | 5.77 $\pm$0.07 | 5.58 $\pm$0.03 | 5.65 $\pm$0.14 | 5.63 $\pm$0.07 | 5.55 $\pm$0.03 | 5.53 $\pm$0.07 | 5.54 $\pm$0.09 | 5.51 $\pm$0.05 | 5.66 $\pm$0.17 |
| INDEQS$_{outer}$ | 5.33 $\pm$0.10 | 5.33 $\pm$0.11 | 5.31 $\pm$0.12 | 5.34 $\pm$0.13 | 5.28 $\pm$0.14 | 5.35 $\pm$0.09 | 5.27 $\pm$0.18 | 5.24 $\pm$0.18 | 5.23 $\pm$0.14 | 5.27 $\pm$0.14 | 5.23 $\pm$0.12 | **5.20** $\pm$0.12 |
| INDEQS$_{inner}$ | 5.80 $\pm$0.06 | 5.77 $\pm$0.03 | 5.63 $\pm$0.04 | 5.62 $\pm$0.06 | 5.51 $\pm$0.10 | 5.54 $\pm$0.09 | 5.44 $\pm$0.04 | 5.37 $\pm$0.06 | 5.43 $\pm$0.04 | 5.39 $\pm$0.12 | 5.38 $\pm$0.10 | 5.34 $\pm$0.07 |
| AGC - Informed | 5.33 $\pm$0.03 | 5.33 $\pm$0.02 | 5.33 $\pm$0.06 | 5.33 $\pm$0.05 | 5.32 $\pm$0.05 | 5.29 $\pm$0.02 | 5.26 $\pm$0.02 | 5.29 $\pm$0.04 | 5.33 $\pm$0.03 | 5.28 $\pm$0.07 | 5.29 $\pm$0.02 | 5.28 $\pm$0.08 |

Table 9: MAEs for STG-NCDE and INDEQS$_{outer}$ with different prediction decoders on the river discharge task over 5 seeds.

|  | convolutional decoder | NODE$_g$ decoder | NODE$_{ctrl}$ decoder |
|---|---|---|---|
| STG-NCDE | 5.59 $\pm$0.13 | 5.11 $\pm$0.06 | 5.11 $\pm$0.06 |
| INDEQS$_{outer}$ | 5.33 $\pm$0.12 | 5.01 $\pm$0.16 | 5.07 $\pm$0.08 |

Table 10: Overview of MAE, number of parameters, training time and the epoch when validation loss last improved, with standard deviations, for different outer-inner configurations for the **PeMS08 traffic** prediction over 5 runs with different random seeds.

|  | Outer | Inner | MAE (test) | #Params. | MAE (val.) | Train time | Last improvement |
|---|---|---|---|---|---|---|---|
|  | AGC | Identity | 17.85 ±0.40 | 45,152 | 18.51 ±0.48 | 135.49 min ±0.88 | 196.60 ±1.82 |
|  | AGC | Informed | 17.24 ±0.12 | 45,152 | 17.80 ±0.17 | 135.24 min ±1.19 | 175.80 ±30.78 |
| STG-NCDE | Identity | AGC | 16.55 ±0.20 | 45,152 | 17.00 ±0.22 | 129.31 min ±1.04 | 183.00 ±3.94 |
| INDEQS$_{outer}$ | Informed | AGC | **16.50** ±0.19 | 45,152 | 16.88 ±0.28 | 143.18 min ±1.45 | 169.00 ±11.20 |

Table 11: MAE with standard deviation over 5 runs for different outer-inner graph configurations for increasing powers of the adjacency matrix for the **PeMS08 traffic** prediction.

|  | Adjacency matrix power | | | | |
|---|---|---|---|---|---|
|  | 1 | 2 | 3 | 4 | 5 |
| AGC - Informed | 17.24 ±0.11 | 17.22 ±0.30 | 17.13 ±0.18 | 17.27 ±0.21 | 17.23 ±0.23 |
| AGC - Identity | 17.85 ±0.36 | - | - | - | - |
| INDEQS$_{outer}$ | 16.50 ±0.17 | **16.39** ±0.18 | 16.53 ±0.13 | 16.53 ±0.19 | 16.53 ±0.12 |
| STG-NCDE | 16.55 ±0.18 | - | - | - | - |

Table 12: Mean test MAE with standard deviation over 5 runs, the best test MAE of these, number of parameters, mean validation MAE with standard deviation and train time for different decoder architectures for INDEQS$_{outer}$ for the **advection simulation** prediction on a graph with 64 nodes with training for **200 epochs**.

| Decoder type | MAE | best MAE | #Params. | MAE (Val.) | Train time |
|---|---|---|---|---|---|
| convolutional(discrete) | **1.01** ±0.06 | 0.92 | 72,396 | **1.00** ±0.05 | **20.31** min ±0.11 |
| NODE$_g$ (continuous) | 2.81 ±0.51 | 2.28 | 72,033 | 2.76 ±0.53 | 30.91 min ±0.18 |
| NODE$_{ctrl}$ (continuous) | 2.95 ±0.20 | 2.62 | 74,274 | 2.92 ±0.19 | 51.63 min ±1.33 |

Table 13: Mean test MAE with standard deviation over 5 runs, the best test MAE of these, number of parameters, mean validation MAE with standard deviation and train time for different decoder architectures for INDEQS$_{outer}$ for the **advection simulation** prediction on a graph with 64 nodes with training for **400 epochs**.

| Decoder type | MAE | best MAE | #Params. | MAE (Val.) | Train time |
|---|---|---|---|---|---|
| convolutional(discrete) | **0.92** ±0.05 | 0.86 | 72,396 | **0.91** ±0.05 | **41.22** min ±0.13 |
| NODE$_g$ (continuous) | 2.14 ±0.47 | 1.48 | 72,033 | 2.10 ±0.47 | 62.86 min ±0.41 |
| NODE$_{ctrl}$ (continuous) | 2.15 ±0.24 | 1.98 | 74,274 | 2.11 ±0.25 | 104.51 min ±1.30 |

Table 14: Mean test MAE with standard deviation over 5 runs, the best test MAE of these, number of parameters, mean validation MAE with standard deviation and train time for different decoder architectures for INDEQS$_{outer}$ for the **water discharges** prediction.

| Decoder type | MAE | best MAE | #Params. | MAE (Val.) | Train time |
|---|---|---|---|---|---|
| convolutional(discrete) | 5.30 ±0.15 | 5.15 | 415,879 | 5.43 ±0.09 | **41.35** min ±2.05 |
| NODE$_g$ (continuous) | 5.07 ±0.08 | 4.96 | 415,619 | **5.20** ±0.10 | 54.78 min ±1.31 |
| NODE$_{ctrl}$ (continuous) | **5.01** ±0.16 | **4.84** | 416,500 | 5.20 ±0.14 | 92.00 min ±0.84 |

Table 15: Mean test MAE with standard deviation over 5 runs, the best test MAE of these, number of parameters, mean validation MAE with standard deviation and train time for different decoder architectures for INDEQS$_\mathrm{outer}$ for the **PeMS08 traffic** prediction.

| Decoder type | MAE | best MAE | #Params. | MAE (Val.) | Train time |
|---|---|---|---|---|---|
| convolutional(discrete) | 16.50 $\pm$0.19 | 16.25 | 45,152 | 16.88 $\pm$0.28 | **143.72** min $\pm$0.47 |
| NODE$_g$ (continuous) | **16.27** $\pm$0.09 | **16.12** | 44,789 | **16.63** $\pm$0.09 | 220.87 min $\pm$2.85 |
| NODE$_{ctrl}$ (continuous) | 16.37 $\pm$0.12 | 16.20 | 47,030 | 16.75 $\pm$0.21 | 365.49 min $\pm$2.24 |

### A.12 Notational conventions

## Nomenclature

**Advection Simulation**

| | |
|---|---|
| $\delta(e)$ | Edge length |
| $\mathbf{A}_{\mathcal{E}}$ | (Directed) edge transition matrix |
| $\mathbf{v}$ | Velocity vector field |
| $\mathbf{y}$ | Vector field of advected quantity |
| $\Omega_e$ | Continuous edge domain |
| $a_{ij} \in \mathbf{A}_{\mathcal{E}}$ | Matrix elements |
| $p_{ij}$ | Edge transition probability |
| $V_e$ | Function space on edge |

**Neural Differential Equations**

| | |
|---|---|
| $\boldsymbol{f_\theta}, \boldsymbol{g_\gamma}$ | Vector field matrices |
| $\boldsymbol{H}(t), \boldsymbol{Z}(t)$ | Hidden state matrices |
| $\boldsymbol{X}(t)$ | Control matrix |
| $\mathcal{T} = [0, T]$ | Training time interval |
| $\tau$ | Final prediction time point |
| $d_Z, d_H$ | Hidden state matrix dimensions |
| $d_z, d_h$ | Hidden path dimensions (per node) |
| $M$ | Number of prediction timesteps |
| $t \in \mathcal{T}$ | Virtual time |
| $T$ | Final training time point |

**Experimental input**

| | |
|---|---|
| $d_x$ | Dimension of data input (per node) |
| $s$ | Arbitrary parametrization |
| $t_i$ | Observation time points |
| $x(s)$ | Interpolated data input |
| $x_i($ | Data input (discrete) |

**Graph Structure**

| | |
|---|---|
| $\mathcal{E}$ | Set of edges |
| $\mathcal{G}$ | Directed graph |
| $\mathcal{V}$ | Set of vertices/nodes |
| $e \in \mathcal{E}$ | Edge |
| $v \in \mathcal{V}$ | Vertex/node |

**INDEQS specific**

| | |
|---|---|
| $\boldsymbol{A}_{\mathcal{V}}^{\text{inner}}$ | Inner vertex transition matrix |
| $\boldsymbol{A}_{\mathcal{V}}^{\text{outer}}$ | Outer vertex transition matrix |
| $\boldsymbol{E}$ | Node embedding matrix |
| $\boldsymbol{I}$ | Identity matrix |
| $\mathbf{A}_{\mathcal{V}}$ | Vertex adjacency matrix |
| $\mathbf{I}$ | Vertex incidence matrix |
| $B$ | Bias vector |
| $C$ | Node embedding dimension |
| $W$ | Weight transformation matrix |

**Neural Networks (general)**

| | |
|---|---|
| $\hat{\mathbf{Y}}$ | Prediction |
| $\mathbf{Y}$ | Ground truth |
| $\mathcal{L}$ | Loss function |
| $\phi$ | Softmax activation |
| $\sigma$ | Rectified linear unit (ReLU) activation |
| $\theta, \gamma$ | Neural network parameters |
| $d_y$ | Hidden state dimension (per node) |
| $l_\theta(y)$ | Decoder: (convolutional) affine layer |
| $m_\theta(x)$ | Encoder: (linear) affine layer |
| $y$ | Hidden state variable |

### A.13 Comparative overview of Graph Neural Differential Equation methods

In Table 16 and Table 17 we compare INDEQS to Graph Neural Differential Equations methods that are most similar.

Table 16: Comparative overview of Graph Neural Differential Equations methods.

| | INDEQS (this work) | STG-NCDE (Choi et al., 2022) | GDE (Poli et al., 2021) | GN-CDE (Qin et al., 2026) | NDCN (Zang & Wang, 2020) | STGODE (Fang et al., 2021) |
|---|---|---|---|---|---|---|
| Code available | ✔ | ✔ | ✔ | ✗ | ✔ | ✔ |
| Control | Continuous-time node feature control | Continuous-time node feature control | Discrete node feature control via GRU | Continuous-time (graph structure + feature) control | ✗ | ✗ |
| NDE/DE structure | 2 stacked 1st order NCDEs | 2 stacked 1st order NCDEs | 1st/2nd order NODE | 1st order NCDE | 1st order NODE | 2 chained sets of 3 parallel blocks of 1st order NODEs |
| Graph dynamics | Static | Static | Discretely dynamic | Continuously dynamic | Static | Static |
| Graph informed | ✔ | ✗ | ✔ | ✔ | ✔ | ✔ |
| Spatial processing | Adjacency matrix + adaptive graph convolution | Adaptive graph convolution | Graph convolution (laplacian) in vector field | Adjacency matrix as Control | Linear diffusion operator in vector field | 2 adjacency matrices in vector field |
| Edge features processing | ✗ | ✗ | ✗ | ✗ | ✗ | ✗ |
| Spatial domain | Discrete | Discrete | Discrete | Discrete | Discrete | Discrete |
| Temporal architecture | NCDE, convolutional layer | NCDE, convolutional layer | NODE with GRU jumps | NCDE | NODE | TCN (Bai et al., 2018) |
| Irregular data | ✔ | ✔ | ✔ | ✔ | ✔ | ✗ |
| Internal temporal data processing | Continuous encoding on context window, Discrete decoding on horizon | Continuous along NCDE time dimension | Continuous with virtual jumps along NODE time dimension | Continuous along NCDE time dimension | Continuous along NODE time dimension | all-at-once (NODE orthogonal to time dimension) |

Continued on next page

Table 16: Comparative overview of Graph Neural Differential Equations methods. (Continued)

| | INDEQS (this work) | STG-NCDE (Choi et al., 2022) | GDE (Poli et al., 2021) | GN-CDE (Qin et al., 2026) | NDCN (Zang & Wang, 2020) | STGODE (Fang et al., 2021) |
|---|---|---|---|---|---|---|
| Application domains | Hydrology, traffic flow, graph edge advection | Traffic flow | Citation networks, mechanical multi-particle system, traffic speed | Heat diffusion, mutualistic interaction, gene regulation | Heat diffusion, mutualistic interaction, gene regulation | Traffic speed/flow |
| Tasks | Structured sequence forecasting | Structured sequence forecasting | Transductive node classification, Multi–agent trajectory extrapolation, Forecasting | Continuous-time network dynamics prediction, Structured sequence prediction, Node semi-supervised classification | Continuous-time network dynamics prediction, Structured sequence prediction, NDE Node semi-supervised classification | Structured sequence forecasting |

Table 17: Comparative overview of Graph Neural Differential Equations methods.

| | INDEQS (this work) | CG-ODE (Huang et al., 2021) | CF-GODE (Jiang et al., 2023a) | HOPE (Luo et al., 2023) | GRAM-ODE (Liu et al., 2023) | AG-NCDE (Yao & Zhou, 2025) |
|---|---|---|---|---|---|---|
| Code available | ✔ | ✔ | ✗ | ✗ | ✔ | ✗ |
| Control | Continuous-time node feature control | ✗ | ✗ | ✗ | ✗ | Continuous-time node feature control |
| NDE/DE structure | 2 stacked 1st order NCDEs | 2 coupled ODEs (nodes, edges) | ODE parametr. by GNN | 2 coupled 2nd order ODEs (nodes, edges) | 3 coupled blocks of GNN based ODEs (global, local, edge) | 2 stacked 1st-order NCDEs |
| Graph dynamics | Static | Dynamic | Static | Dynamic | Dynamic | Dynamic |
| Graph informed | ✔ | ✔ | ✔ | ✔ | ✔ | ✗ |

Table 17: Comparative overview of Graph Neural Differential Equations methods. (Continued)

| | INDEQS (this work) | CG-ODE (Huang et al., 2021) | CF-GODE (Jiang et al., 2023a) | HOPE (Luo et al., 2023) | GRAM-ODE (Liu et al., 2023) | AG-NCDE (Yao & Zhou, 2025) |
|---|---|---|---|---|---|---|
| Spatial processing | Adjacency matrix + adaptive graph convolution | GNN in initial encoder | GNN in vector field | Spatial + spectral convolution | Adjacency (connection map + DTW graph) + dynamic semantic edges | NCDE + inferred, dynamic adjacency Matrix + graph convolution |
| Edge features processing | ✗ | ✔ | ✔ | ✔ | ✔ | ✗ |
| Spatial domain | Discrete | Discrete | Discrete | Discrete | Discrete | Discrete |
| Temporal architecture | NCDE, convolutional layer | GNN + temp. self attention | NODE | Spatial + spectral convolution + temp. self attention | TCN + multi ODE-GNN + multi-head attention | NCDE + external attention on two time scales |
| Irregular data | ✔ | ✔ | ✗ | ✗ | ✗ | ✔ |
| Internal temporal data processing | Continuous encoding on context window, discrete decoding on horizon | Discrete encoding along context window, continuous decoding along horizon window | Continuous decoding via NODE | Discrete encoding, continuous decoding | Discrete encoding via TCN, continuous processing via ODE-GNNs, discrete decoding on horizon | Continuous encoding on context window, discrete decoding on horizon |
| Application domains | Hydrology, traffic flow, graph edge advection | Disease spread | Disease control | Disease spread, Opinion migration on social networks, Spring oscillation | Traffic flow | Traffic flow |
| Tasks | Structured sequence forecasting | Forecasting | Causal inference, counterfactual outcomes | Sequence representation Learning | Forecasting | Forecasting |

### A.14 River network maps

The river topology of the river discharge task is depicted in Figure 14.

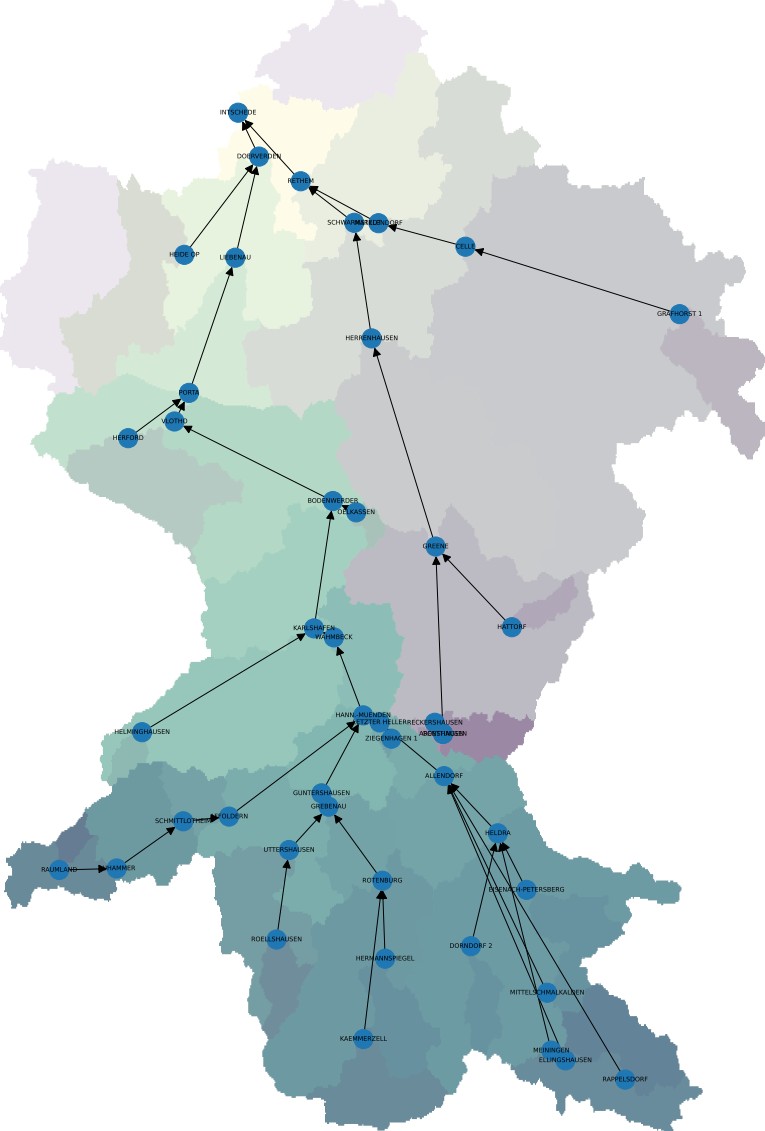

Figure 14: The graph network of the discharge stations of the Weser river and their catchments

