# OpenReview forum: "INDEQS: Informed Neural controlled Differential EQuationS"
_TMLR — Under review for TMLR_

### Review · Reviewer_pHVx · 2026-07-10

**Summary Of Contributions:**

This paper largely follows https://ojs.aaai.org/index.php/AAAI/article/view/20587 (Choi et al., 2022) which proposes to use Neural Controlled Differential Equations to model continuous spatio-temporal data, such as for traffic forecasting. Instead of learning a soft adjacency matrix of the spatio graph as proposed in Choi et al. (2022), this work proposes to explicitly inject the adjacency matrix into the model to strengthen the prior knowledge.

This paper is not very clearly written, so I'm not completely sure about the difference compared to Choi et al. (2022). I feel that the overall framework is already fully covered by Choi et al. (2022), while this work only adds some small tweaks that might be special cases of Choi et al. (2022). Let me lay down the notations first for further discussion.

A Neural Controlled Differential Equation is like a continuous Recurrent Neural Network. The training data is a time-series $x(t)$, which can be the (scalar) traffic volume of a specific location (at time $t$) in our example. As in RNN, one assumes that $x(t)$ can be explained by the evolution of a hidden state $\boldsymbol{h}(t)$ of $D$ dimension, and at each "time-step", the model takes $\boldsymbol{h}(t)$ and $x(t)$ as input and tries to predict the "next" hidden state. In a continuous setting, this is modeled by a differential equation:

$$
\frac{\mathrm{d}\boldsymbol{h}(t)}{\mathrm{d}t}=f(\boldsymbol{h}(t);\boldsymbol{\theta})\frac{\mathrm{d}x(t)}{\mathrm{d}t}
$$

where $f(\boldsymbol{h}(t);\boldsymbol{\theta})$ is a deep neural network with trainable parameter $\boldsymbol{\theta}$ and input $\boldsymbol{h}(t)$.

One can also convert the training data $x(t)$ into a vector $\boldsymbol{x}(t)$; one way of doing it is to define $\boldsymbol{x}(t):=(x(t+1),\ldots, x(t+N))$ where $x(1),\ldots,x(N)$ are the actually observed data points while a general $x(t)$ is given by an interpolation. The reason for this vectorization is because one has to provide $\boldsymbol{h}(0)$ in order to solve the differential equation; for which one usually sets $\boldsymbol{h}(0):=\tilde{f}(\boldsymbol{x}(0);\tilde{\boldsymbol{\theta}})$ where $\tilde{f}$ is a trainable deep neural network similar to $f$. -- This way, the initial hidden state $\boldsymbol{h}(0)$ can see **all** the observed data points.

Then, the deep neural networks $f$ and $\tilde{f}$ are trained by projecting $\boldsymbol{h}(t)$ to a scalar (where the projection matrix is also trainable) and makes it close to the data $x(t)$.

While Choi et al. (2022) is probably doing the vectorization of $\boldsymbol{x}(t)$, it is unclear to me if this work is also doing the same.

Then, when there are multiple locations (i.e., spatio-temporal data), one can stack the $V$ locations together to define a system of differential equations

$$
\frac{\mathrm{d}\boldsymbol{H}(t)}{\mathrm{d}t}=f(\boldsymbol{H}(t);\boldsymbol{\theta})\frac{\mathrm{d}X(t)}{\mathrm{d}t}
$$

where the batched-matrix-multiplication between $f(\boldsymbol{H}(t);\boldsymbol{\theta})$ and $\frac{\mathrm{d}X(t)}{\mathrm{d}t}$ is reducing the $N$-dimensional data-vectorization channels (NOTE: The $V$-dimensional spatio channels are **batched**, not reduced). The resulted $\boldsymbol{H}(t)$ is a $V\times D$ tensor.

In other words, the stacked $\boldsymbol{H}(t)$ simply models $\boldsymbol{h}(t)$ at each location independently. In order to model spatio correlation, Choi et al. (2022) proposes to feed $\boldsymbol{H}(t)$ to a second Neural Controlled Differential Equation (as analogous to a 2-layer RNN):

$$
\frac{\mathrm{d}\boldsymbol{Z}(t)}{\mathrm{d}t}=g(\boldsymbol{Z}(t);\boldsymbol{\theta})f(\boldsymbol{H}(t);\boldsymbol{\theta})\frac{\mathrm{d}X(t)}{\mathrm{d}t}
$$

where the batched-matrix-multiplication between $g(\boldsymbol{Z}(t);\boldsymbol{\theta})$ and $f(\boldsymbol{H}(t);\boldsymbol{\theta})\frac{\mathrm{d}X(t)}{\mathrm{d}t}$ is reducing the $V$-dimensional spatio channels (whereas the $D$-dimensional hidden state is batched). The resulted $\boldsymbol{Z}(t)$ is of $(V\times D)$-dimension, i.e., a stack of $D$-dimensional hidden vectors at each of the $V$ locations. Therefore, $g(\boldsymbol{Z}(t);\boldsymbol{\theta})$ is a $V\times V\times D$ tensor, whereas $f(\boldsymbol{H}(t);\boldsymbol{\theta})\frac{\mathrm{d}X(t)}{\mathrm{d}t}$ is a $V\times D$ tensor.

Choi et al. (2022) is modeling the $V\times V\times D$ tensor $g(\boldsymbol{Z}(t);\boldsymbol{\theta})$ as a $V\times D$ tensor, such that the $V$-dimensional slices are put on the diagonals of $(V\times V)$. Hence, the batched-matrix-multiplication between $g(\boldsymbol{Z}(t);\boldsymbol{\theta})$ and $f(\boldsymbol{H}(t);\boldsymbol{\theta})\frac{\mathrm{d}X(t)}{\mathrm{d}t}$ in Choi et al. (2022) is actually equivalent to an element-wise multiplication.

In contrast, the `INDEQS_outer` method proposed in this work is probably injecting the constant adjacent matrix of the spatio graph (which is of $V\times V$ dimension) into the $V\times V\times D$ tensor, instead of simply modeling the diagonal.

It is noteworthy that Choi et al. (2022) does incorporate the adjacency of the spatio graph implicitly. This is done inside the definition of $g(\boldsymbol{Z}(t);\boldsymbol{\theta})$, where the $V$ slices of $\boldsymbol{Z}(t)$ gets mixed as similar to an attention mechanism. The "attention probability" in this mechanism, as defined by the $V\times V$ matrix $\phi(\sigma(\boldsymbol{E} \boldsymbol{E}^{\top}))$ in Choi et al. (2022), (where $\phi$ is softmax, $\sigma$ is ReLU, and $\boldsymbol{E}$ is the stack of trainable per-location embeddings,) can be injected with priors coming from the adjacent matrix of the spatio graph. Although Choi et al. (2022) did not explicitly do this -- it simply trains the attention without any prior -- their paper clearly states that $\phi(\sigma(\boldsymbol{E} \boldsymbol{E}^{\top}))$ can be conceptually viewed as corresponding to the normalized adjacency matrix.

The `INDEQS_inner` method proposed in this work is similar to the previous idea. However, for some reason the authors propose to completely replace the learned attention matrix by the constant adjacency matrix. This might work in some problems with strong priors and scarce data, but I don't see why the method should be positioned as a general way to incorporate prior knowledge from spatio graphs.

**Audience:**

Yes

**Audience Explanation:**

The task of modeling spatio-temporal is important, and Neural Controlled Differential Equations is a promising and intriguing approach. TMLR's audience would be interested in knowing the findings of this paper once it is precisely formulated.

**Broader Impact Concerns:**

None.

**Claims And Evidence:**

No

**Claims Explanation:**

The proposed methods probably only work when one has a directed graph that completely explains all the causal relations. In such cases, I wouldn't be surprised if a specific modeling which heavily uses the graph information could be better.

However, I believe most of the graphs in practice are not like that -- for example, what if there are loops in the graphs? What if the graph is undirected, representing correlation rather than causality? What if the graph only covers partial information -- for example, in traffic flow forecasting, a graph of highway networks probably won't cover all the exits, so there could be traffic going in and out of the networks, everywhere at every moment? Currently, the narration of the paper seems to claim a general framework to _"inject prior information about an underlying directed graph into the vector field of a Neural Controlled Differential Equation (NCDE) to infer the future dynamics at the vertices."_ -- Which I don't think is precisely covering the exact contribution of this work.

**Requested Changes:**

The paper writing needs to be improved a lot. Currently, I can only get a rough idea of what this work is doing, by looking at the formulas and comparing them with Choi et al. (2022) (which is much clearer). Most of the text explaining the formulas in this paper is unreadable.

Especially at the beginning of Section 4, it is supposed to explain the intuition of the ideas -- I cannot understand it at all.

It might help to use a running example to give concrete shapes of all newly defined tensors. It might also help to make more analogies to RNNs. For high rank tensors, it might be difficult to precisely describe the calculation by common mathematical notations; I think it would be OK to directly write the PyTorch ops in such cases -- people would understand.

---

> ### Author Response · Authors · 2026-07-22
> **Author rebuttal I**
>
> We thank the reviewers for their feedback and for their evaluation of our work. We appreciate that the reviewers find the results to be of interest to the TMLR audience. Below, we address each concern in turn. The corresponding clarifications and changes have been incorporated into the revised manuscript; all modifications are marked in blue.
>
> ## Clarity/Presentation
> We acknowledge that the original submission was formula-centric in the background and method section for INDEQS. In the revision, we have reworked parts of these sections to improve the narrative flow and reduce the burden on the reader. Concretely, we now:
> - Add explicit sentences explaining each non-trivial formula, rather than relying purely on notation.
> - Use terminology suggested by the reviewer (e.g., "batched") to describe tensor operations at a higher level.
>
> We hope that these changes substantially improve the clarity and presentation of the work, while keeping the mathematical core intact.
>
> ## Differentiation to STG-NCDE
> We appreciate the concern that the overall framework might appear to be already covered by Choi et al. (2022) (STG-NCDE). We clarified in the paper that INDEQS builds on STG-NCDE but modifies it in a principled way to incorporate known prior graph information.
>
> Choi et al. (2022) lay out a framework for Neural Controlled Differential Equations on spatio-temporal graphs, but they do not investigate graph informedness. Their model relies on a learned, data-adaptive graph embedding. Our analysis on advection simulation data shows that, in this setting, the learned embedding alone is not capable of recovering the future behavior of the dynamics at the graph nodes from time series data. This raises the question of whether additional prior graph structure can alleviate this limitation and how such structure should be introduced.
>
> INDEQS answers this question by:
>
> - Introducing mechanisms (INDEQS_inner and INDEQS_outer) that inject prior information from a directed graph representing the underlying process into the NCDE vector fields, in addition to the learned graph convolution of STG-NCDE.
> - Systematically studying different positions at which the graph can enter the model: inside the vector field (inner) versus between the vector field and the control (outer).
> - Conducting extensive experiments in a controlled setting (graph advection simulation) and in a real-world setting (river discharge forecasting), to quantify how graph informedness affects predictive performance.
>
> These clarifications, together with the new experiments and analyses, make the distinction between STG-NCDE and INDEQS explicit: INDEQS is a targeted modification of STG-NCDE that introduces and studies prior graph information.
>
> ## Clarification of Vectorization
>
> The reviewer refers in several places to a "vectorization" of the training data of the form $\boldsymbol{x}(t) := (x(t+1), \ldots, x(t+N))$, used to construct an initial hidden state $(\boldsymbol{h}(0) = \tilde{f}(\boldsymbol{x}(0); \tilde{\boldsymbol{\theta}}))$ that "can see all the observed data points". This vectorization is not part of our INDEQS framework, nor, to our understanding, of the original STG-NCDE formulation.
>
> In Neural Controlled Differential Equations (see section 3.2), it is not necessary to vectorize the entire time series at $(t=0)$. Instead, later data points $(\{x(t) \mid t > t_0\})$ are encoded via the control term in the integral over the temporal context window. The initial hidden state $(\boldsymbol{h}(t_0))$ is constructed from the data point $(x(t_0))$ via $\tilde{f}(x(t_0))$, which can be viewed as a projection into a higher-dimensional hidden space. Only through integration against $\mathrm{d}x(t)$ are subsequent data points incorporated into $\boldsymbol{h}(t)$. After integration up to time $T$, the hidden state has "seen" the full path $\{x(t) \mid t \in [t_0, T]\}$.

---

> ### Author Response · Authors · 2026-07-22
> **Author rebuttal II**
>
> ## Dimension Corrections
>
> In the reviewer's summary of contributions, we would like to clarify the tensor dimensions, which our original notation unfortunately obscured — the codomains of $\boldsymbol{g}$ and $\boldsymbol{f}$ were misstated in Section 3.3, which likely contributed to the confusion; we have corrected this in the revised manuscript.
>
> >$\frac{\mathrm{d}\boldsymbol{Z}(t)}{\mathrm{d}t}=g(\boldsymbol{Z}(t);\boldsymbol{\theta})f(\boldsymbol{H}(t);\boldsymbol{\theta})\frac{\mathrm{d}X(t)}{\mathrm{d}t}$
> where the batched-matrix-multiplication between $g(\boldsymbol{Z}(t);\boldsymbol{\theta})$ and $f(\boldsymbol{H}(t);\boldsymbol{\theta})\frac{\mathrm{d}X(t)}{\mathrm{d}t}$
> is reducing the $V$-dimensional spatio channels (whereas the $D$-dimensional hidden state is batched). The resulted $\boldsymbol{Z}(t)$ is of $(V \times D)$-dimension, i.e., a stack of $D$-dimensional hidden vectors at each of the locations. Therefore, $g(\boldsymbol{Z}(t);\boldsymbol{\theta})$ is a $V \times V \times D$ tensor, whereas $f(\boldsymbol{H}(t);\boldsymbol{\theta})\frac{\mathrm{d}X(t)}{\mathrm{d}t}$
> is a $V \times D$ tensor
>
> The dimensions differ from this reading in one respect: we distinguish between $d_h$ and $d_z$ for the hidden state dimensions at each node: $\boldsymbol{g}$ is a $V \times d_z \times d_h$ tensor and $f(\boldsymbol{H}(t);\boldsymbol{\theta})\frac{\mathrm{d}X(t)}{\mathrm{d}t}$ is a $V \times d_h$ tensor. Their product is a $V \times d_z$ tensor --i.e., the contraction is over the hidden dimension at each node, while the node dimension $V$ is batched
>
> ## Graphs in Practice and Perturbed Graph Analysis
>
> The reviewer rightly points out that graphs used in practice are often imperfect and may only partially capture the true causal structure. We agree and did already highlighted this in the limitations of the discussion section, noting that our approach assumes access to a reasonably accurate underlying graph and that our evaluation covers only a limited set of graph types, sizes, and domains.
>
> To address this concern further, we have added an experiment specifically designed to test the robustness of INDEQS to graph misspecification (Appendix A9.2; Figure 11 shows the full, more fine-grained grid over the percentage of removed/added edges) Starting from the ground-truth directed graph derived from the underlying physical process, we perturb it by removing or adding edges and compare INDEQS under these perturbed graphs against both the unperturbed graph (the 0% row) and the uninformed STG-NCDE baseline. An abridged version
>
> | Perturbation | STG-NCDE | $\text{INDEQS}_{\text{inner}}$ | $\text{INDEQS}_{\text{outer}}$ |
> |---|---|---|---|
> | −100% | 5.64 ± 0.1174 | — | 5.64 ± 0.1174 |
> | −60% | — | 4.89 ± 0.1079 | 4.19 ± 0.0513 |
> | −5% | — | 1.87 ± 0.2473 | 1.34 ± 0.0610 |
> | 0 | 5.64 ± 0.1174 | 1.44 ± 0.2252 | 1.01 ± 0.0563 |
> | +5% | — | 1.53 ± 0.1287 | 1.11 ± 0.0689 |
> | +60% | — | 2.83 ± 0.1861 | 1.70 ± 0.1043 |
> | undirected | — | 2.82 ± 0.2590 | 1.84 ± 0.0789 |
> | transposed | — | 6.17 ± 0.0532 | 5.23 ± 0.1427 |
>
> INDEQS_outer with a perturbed graph consistently outperforms the uninformed baseline, and INDEQS_inner does so in all but the transposed case. At −100%, no prior edges remain and INDEQS_outer reduces to the uninformed baseline, as expected. The transposed graph marks the adversarial extreme: INDEQS_inner falls below the baseline there, while INDEQS_outer still performs slightly better. This analysis supports the claim that INDEQS can leverage partially correct prior graphs and that the graph informedness is useful even when the available graph does not perfectly encode all causal relations.
>
> This analysis shows that INDEQS can leverage partially correct prior graphs and that the graph informedness is useful even when the available graph does not perfectly encode all causal relations.
>
> ## Clarification of the Scope of Our Framework
>
> The reviewer notes that our narration might be interpreted as claiming a general framework to "inject prior information about an underlying directed graph into the vector field of an NCDE to infer the future dynamics at the vertices", which the reviewer feels do not precisely match our actual contribution.
>
> We do not claim a fully general framework for all graphs in practice; instead, we present and study a concrete class of directed graph-informed NCDE models suited to processes with reasonably informative directed graphs.
>
> INDEQS proposes specific mechanisms for incorporating prior information from directed graphs into NCDE-based spatio-temporal models, focusing on settings where the directed graph is believed to approximate the main flow of information or mass (e.g., advection, river networks).
>
> Our experiments illustrate these mechanisms on controlled advection simulations and river discharge forecasting; they do not establish full generality across all possible graph types (e.g., arbitrary undirected correlation graphs or graphs with severe coverage gaps).

---

> > ### Author Response · Authors · 2026-07-22
> > **Author rebuttal III**
> >
> > We hope that these changes, together with the additional experiments and clarifications above, address the reviewer's concerns and improve the clarity of the work. We would be happy to elaborate on any remaining points during the discussion period, and kindly invite the reviewer to reconsider the evaluation of our work.

---

### Review · Reviewer_cjFY · 2026-07-11

**Summary Of Contributions:**

This work proposes a new graph neural controlled differential equations framework, named Informed Neural Controlled Differential Equations (INDEQS). By directly encoding the connectivity information as an unlearnable fixed linear layer in the existing graph neural controlled differential equations framework, the INDEQS models are by construction more faithful to the known graph information, hence shows potential to yield better predictions over the existing approaches. It is a simple and computationally inexpensive fix to the previous graph neural controlled differential equations framework that can enhance performance.

**Additional Comments:**

The authors acknowledges that the ground truth graph information might not be always known a priori.
So, there should always be a possibility that *$\boldsymbol{A}_{\mathcal{V}}$* we hardcode inside INDEQS poorly represents the ground truth information of the underlying graph.
In that case, it feels like the performance of INDEQS would be worse, and the existence of *$\boldsymbol{A}_{\mathcal{V}}$*, being fixed inside the model, would make the training worse.
I would like to hear from the authors about how they think about this potential issue.

For *$\boldsymbol{A}_{\mathcal{V}}(k)$* in equation (8), wouldn't the unnormalized sum (as in the current submission) be problematic if *$\|\boldsymbol{A}_{\mathcal{V}}\| \geq 1$*, having an exponentially growing matrix norm with respect to $k$?

As a side note, due to the lack of my background in the field of graph neural controlled differential equations, I am not completely certain whether people working in that field would actually be interested in this work.

**Audience:**

Yes

**Audience Explanation:**

The proposed technique is a simple addition to the existing method. I guess a method that can potentially perform better with very low additional cost would be interesting to the practitioners that work on that field of graph neural controlled differential equations.

**Broader Impact Concerns:**

This work proposes a general modeling technique of graph neural controlled differential equations.
Thus, I don't think there are any further broader impact concerns specific to this submission.

**Claims And Evidence:**

Yes

**Claims Explanation:**

The claims of the submission are mainly about empirical results, suggesting a new modeling technique. Codes used in the experiments are provided, backing up the claims. The high-level idea that the model would perform better if we force it to exploit the known graph information rather than learn it from scratch is sufficiently convincing.

**Requested Changes:**

1. Shouldn't the codomain of *$\boldsymbol{g}_\boldsymbol{\gamma}$* be *$\mathbb{R}^{(|\mathcal{V}|\times d_z) \times (|\mathcal{V}|\times d_h)}$*, instead of *$\mathbb{R}^{|\mathcal{V}|\times d_z} \times \mathbb{R}^{|\mathcal{V}|\times d_h}$*? Similar goes to *$\boldsymbol{f}_\boldsymbol{\theta}$*. Also, it should be clarified whether the authors are using $|\mathcal{V}|\times d_h$-dimensional space as a space of matrices or a space of vectors. More precisely, in appendix A.7.1, the integrand in equation (18) looks like it is a matrix-vector product, but right beneath that equation it says that $\boldsymbol{1}$ is a *matrix* of ones. Then the integrand must be written as a tensor product, or from the beginning we should have considered $\mathbb{R}^{|\mathcal{V}|\times d_h}$ as a space of ${(|\mathcal{V}| d_h)}$-dimensional "flattened" vectors.
2. The notations are often inconsistent, for example, in (1) we should have $\boldsymbol{f}$ instead of $f$, in the first line of appendix A.4 the vertex adjacency matrix is incorrectly denoted by $\mathbf{A}_{\mathcal{E}}$, and after defining the matrix notations the edge transition matrix is now denoted by $\mathbf{I}$ instead of $\mathbf{A}$. Also, while not technically incorrect, in section 3.3 the time series is denoted by $\mathbf{Y}$ but the control is denoted by $\mathbf{X}$, unlike the previous section 3.2.
3. In equation (4), it would be more informative to write $\boldsymbol{g}_{\boldsymbol{\gamma}}(\cdot)$ rather than specifying its argument $\boldsymbol{Z}(t)$.
4. The caption of Figure 2 uses the notation *$\boldsymbol{A}_{\mathcal{V}}$*, which has not yet been defined. Also, $\boldsymbol{A}$ should be in boldface.
5. While the core claims and results do not strictly rely on the materials stated in appendix A.4, as long as the authors decided to include it, I wish that part is rewritten to enhance clarity. In addition to the typos mentioned above, the first sentence of that section indicates that it is going to be on how the edge transition matrix can be obtained from the vertex adjacency matrix, but it only explains how the former can be obtained from the vertex *incidence* matrix.
6. There are several formatting errors in the citations (mainly incorrectly positioned parentheses), please fix all of them.

---

> ### Author Response · Authors · 2026-07-22
> **Author rebuttal I**
>
> We thank the reviewer for the careful reading of our paper and the positive assessment of our empirical evidence, our provided code and the potential interest of INDEQS for the community. Below we address each of the points raised. The corresponding clarifications and changes have been incorporated into the revised manuscript; all modifications are marked in blue.
>
> ### Notation and codomain dimensions
> > 1. Shouldn't the codomain of $\boldsymbol{g}_{\gamma}$ be $\mathbb{R}^{(|\mathcal{V}|\times d_z)\times (|\mathcal{V}|\times d_h})$
>
> We thank the reviewer also for carefully checking the dimensionalities in our description. We agree that the codomain of $\boldsymbol{g}$ (and similarly $\boldsymbol{f}$ ) was misstated. In the revised version we correct it to $\mathbb{R}^{|\mathcal{V}|\times d_z \times d_h}$. We have corrected this in the revised manuscript, harmonizing the notation across Sections 3 and 4 and the appendix. The implementation already used the correct shapes, so this was a purely presentational error rather than a conceptual flaw.  Further we explicitly clarify for every equation how the tensor multiplication has to be understood. In Appendix A.7.1 we now explicitly state that $\boldsymbol{g}{\boldsymbol{\gamma}}(P(t'))\boldsymbol{1}$ is a batched matrix-vector product over the node dimension.
>
> > 2. The notations are often inconsistent
>
> We further accepted and implemented all the reviewer's notational comments, fixing the adjacency/transition matrix symbols in Appendix A.4, and making the notation in Sections 3.2–3.3 consistent.
>
> > 4. The caption of Figure 2 uses the notation $\boldsymbol{A}_{\mathcal{V}}$
>
> We have revised the caption of Figure 2 to remove undefined notation.
>
> > 5. While the core claims and results do not strictly rely on the materials stated in appendix A.4 [...] I wish that part is rewritten to enhance clarity.
>
> We rewrote Appendix A.4 to improve clarity and align the first sentence with the actual content. The section now explicitly explains how the edge transition matrix can be obtained from the vertex adjacency matrix (not only from the incidence matrix) and includes a concrete example with graphs and concrete matrices illustrating this construction step‑by‑step.
>
> > 6. There are several formatting errors in the citations
>
> We thank the reviewer for pointing this out and have corrected the citation formatting throughout the revised manuscript.
>
> ### Additional comments – misspecified graph and training stability.
> We appreciate the thoughtful concern that a hard‑coded $\boldsymbol{A}_{\mathcal{V}}$ might poorly represent the true underlying graph in some applications and could in principle hurt performance or optimization. In the INDEQS_outer setting, however, the learned vector fields and adaptive graph convolution provide a mechanism to compensate for such misspecification: during training, the model can downweight or effectively nullify the influence of the fixed adjacency (e.g., by setting the corresponding weights to zero) and rely more on the adaptive component.
>
> To address this concern further, we have added an experiment specifically designed to test the robustness of INDEQS to graph misspecification (Appendix A9.2; Figure 11 shows the full, more fine-grained grid over the percentage of removed/added edges) Starting from the ground-truth directed graph derived from the underlying physical process, we perturb it by removing or adding edges and compare INDEQS under these perturbed graphs against both the unperturbed graph (the 0% row) and the uninformed STG-NCDE baseline. An abridged version
>
> | Perturbation | STG-NCDE | $\text{INDEQS}_{\text{inner}}$ | $\text{INDEQS}_{\text{outer}}$ |
> |---|---|---|---|
> | −100% | 5.64 ± 0.1174 | — | 5.64 ± 0.1174 |
> | −60% | — | 4.89 ± 0.1079 | 4.19 ± 0.0513 |
> | −5% | — | 1.87 ± 0.2473 | 1.34 ± 0.0610 |
> | 0 | 5.64 ± 0.1174 | 1.44 ± 0.2252 | 1.01 ± 0.0563 |
> | +5% | — | 1.53 ± 0.1287 | 1.11 ± 0.0689 |
> | +60% | — | 2.83 ± 0.1861 | 1.70 ± 0.1043 |
> | undirected | — | 2.82 ± 0.2590 | 1.84 ± 0.0789 |
> | transposed | — | 6.17 ± 0.0532 | 5.23 ± 0.1427 |
>
> INDEQS_outer with a perturbed graph consistently outperforms the uninformed baseline, and INDEQS_inner does so in all but the transposed case. At −100%, no prior edges remain and INDEQS_outer reduces to the uninformed baseline, as expected. The transposed graph marks the adversarial extreme: INDEQS_inner falls below the baseline there, while INDEQS_outer still performs slightly better. This analysis supports the claim that INDEQS can leverage partially correct prior graphs and that the graph informedness is useful even when the available graph does not perfectly encode all causal relations.

---

> ### Author Response · Authors · 2026-07-22
> **Author rebuttal II**
>
> ### Additional comments – sum of adjacency powers and matrix norms.
>
> Regarding the unnormalized sum $\boldsymbol{A}{\mathcal{V}}(k)$  in Eq. (8), our setting avoids exponential growth of the matrix norm in practice. For directed acyclic graphs (as in the advection simulation and river discharge tasks), there exists a finite $L$ such that $\boldsymbol{A}_{\mathcal{V}}^{L+1}=0$, so the sum over powers only grows up till $L$; in our
> experiments we cap (k) at 12 precisely to stay within this regime. For the undirected traffic flow graph, edge weights are normalized via a Gaussian kernel to lie in ([0,1]), which controls the spectral radius and prevents norm explosion. For very densely connected graphs the empirical benefit of including many higher‑order powers is limited, so in practice we restrict (k).
>
> We thank the reviewer for the constructive feedback. In response, we have added several new experiments and analyses and revised the manuscript throughout. We hope these revisions address the concerns raised, and we would be happy to elaborate on any remaining points during the discussion period.

---

### Review · Reviewer_B5Xr · 2026-07-14

**Summary Of Contributions:**

The paper proposes INDEQS, a graph-informed Neural Controlled Differential Equation model for spatio-temporal forecasting. Its main idea is to inject a known directed adjacency matrix at two different positions in the NCDE: an inner variant that strictly constrains node communication, and an outer variant that combines the prior graph with learnable dependencies. The authors also introduce a continuous advection simulation on graphs and evaluate the method on synthetic data, river discharge forecasting, and PeMS08 traffic prediction. Results show that graph informedness generally improves over the uninformed STG-NCDE baseline, especially on larger graphs, while the inner variant is more parameter-efficient. The main strengths are the clear architectural comparison and controlled synthetic benchmark; the main limitation is that the methodological novelty is fairly incremental and the gains on some real-world tasks are modest.

**Additional Comments:**

N/A

**Audience:**

Yes

**Audience Explanation:**

The paper should interest researchers working on neural differential equations, graph-based time-series forecasting, and physics-informed learning. Its main finding about known graph structure can improve NCDE forecasting, especially for larger graphs and physically meaningful flow networks.

**Claims And Evidence:**

Yes

**Claims Explanation:**

The main claims are supported by experiments on a controlled advection simulation and two real-world datasets, with results reported over multiple random seeds. The analyses of graph size, spatial resolution, adjacency order, decoder choice, efficiency, and parameter count support the claim that incorporating a known graph can improve STG-NCDE, particularly for larger or physically structured graphs. However, the evidence supports this relatively narrow claim rather than broad superiority: the improvement on PeMS08 is small, and INDEQS does not outperform the strongest discrete-time baseline on river forecasting.

**Requested Changes:**

### main

1. **Clarify the novelty and positioning relative to STG-NCDE.** The paper should more explicitly distinguish whether the main contribution is a new model, a graph-informed architectural modification, or an empirical study of where graph priors should enter an NCDE.

2. **Provide stronger evidence for the real-world improvements.** The gain on PeMS08 is very small, so the authors should report statistical significance or confidence intervals for the pairwise differences and discuss whether the improvement is practically meaningful.

3. **Evaluate robustness to inaccurate graph priors.** The current experiments mainly use correct or physically motivated graphs. An ablation with removed, added, or reversed edges would show when informedness helps and when an incorrect graph can hurt.

### minor

4. **Expand comparisons with graph-informed continuous-time baselines.** Additional closely related graph neural ODE/CDE methods, or a clearer explanation of why they are not directly applicable, would improve the empirical positioning.

5. **Separate the effect of the informed encoder from the decoder.** A more systematic matched comparison of convolutional and continuous decoders across all model variants would clarify which gains come from graph informedness and which come from the more expressive decoder.

---

> ### Author Response · Authors · 2026-07-22
> **Author rebuttal I**
>
> We thank the reviewer for the thoughtful and constructive feedback. Below we address each of the requested changes in detail. The corresponding clarifications and changes have been incorporated into the revised manuscript; all modifications are marked in blue.
>
> > 1. Clarify the novelty and positioning relative to STG-NCDE.
>
> We agree this should be stated more explicitly. We now state that INDEQS is a graph-informed architectural modification of STG-NCDE. The main contribution is to inject a known directed adjacency matrix at two distinct locations (inner and outer variants) and to empirically study where such priors should enter an NCDE. We have revised the contribution list at the end of the introduction to make this positioning explicit. The revised contributions are:
> * We introduce a modification to the graph NCDE method STG-NCDE, called
> INDEQS, that incorporates prior directed-graph information into the vector field at two architectural positions: inner (mixing across nodes’ states) and outer (mixing between hidden state and control), yielding a parameter-efficient variant that strictly respects the known adjacency and a more expressive, data-adaptive variant.
> * We propose a continuous-time advection simulation on directed graphs, which produces synthetic spatiotemporal datasets with known ground-truth flow structure and enables controlled study of when graph informedness benefits forecasting.
> * Across synthetic and a real-world benchmark (river discharge), outer informedness reduces MAE relative to an uninformed NCDE at matched parameter count, with the largest gains on larger graphs; inner informedness offers a parameter-efficient alternative that strictly respects the known adjacency.
>
> > 2. Provide stronger evidence for the real-world improvements.
>
> We have performed a Wilcoxon signed-rank test on the pairwise differences between INDEQS_outer and STG-NCDE over 5 independent runs (A.8 Statistical tests); the test does not indicate a statistically significant improvement. STG-NCDE and  INDEQS_outer yields $p = 0.563$, while the comparison with $\text{INDEQS}_{\text{outer}}(k=2)$ yields $p = 0.625$. In line with this, we have revised the main text: instead of stating that INDEQS "outperforms" the baseline on PeMS08, we now describe it as performing comparably, while emphasizing the clearer gains observed on larger and physically structured graphs (advection, river discharge), where the directed adjacency seems to carry genuine flow information.
>
> > 3. Evaluate robustness to inaccurate graph priors.
>
> We thank the reviewer for this thoughtful suggestion. We have added an experiment on the graph advection data specifically designed to test misspecification (Appendix A9.2; Figure 11 shows the full, more fine-grained grid over the percentage of removed/added edges). Starting from the ground-truth directed graph derived from the underlying physical process, we perturb it by removing or adding edges and compare INDEQS under these perturbed graphs against both the unperturbed graph (the 0% row) and the uninformed STG-NCDE baseline. An abridged version
>
> | Perturbation | STG-NCDE | $\text{INDEQS}_{\text{inner}}$ | $\text{INDEQS}_{\text{outer}}$ |
> |---|---|---|---|
> | −100% | 5.64 ± 0.1174 | — | 5.64 ± 0.1174 |
> | −60% | — | 4.89 ± 0.1079 | 4.19 ± 0.0513 |
> | −5% | — | 1.87 ± 0.2473 | 1.34 ± 0.0610 |
> | 0 | 5.64 ± 0.1174 | 1.44 ± 0.2252 | 1.01 ± 0.0563 |
> | +5% | — | 1.53 ± 0.1287 | 1.11 ± 0.0689 |
> | +60% | — | 2.83 ± 0.1861 | 1.70 ± 0.1043 |
> | undirected | — | 2.82 ± 0.2590 | 1.84 ± 0.0789 |
> | transposed | — | 6.17 ± 0.0532 | 5.23 ± 0.1427 |
>
> INDEQS_outer with a perturbed graph consistently outperforms the uninformed baseline, and INDEQS_inner does so in all but the transposed case. At −100%, no prior edges remain and INDEQS_outer reduces to the uninformed baseline, as expected. The transposed graph marks the adversarial extreme: INDEQS_inner falls below the baseline there, while INDEQS_outer still performs slightly better. This analysis supports the claim that INDEQS can leverage partially correct prior graphs and that the graph informedness is useful even when the available graph does not perfectly encode all causal relations.
>
> > 4. Expand comparisons with graph-informed continuous-time baselines.
>
> We expanded the set of continuous-time baselines by adding an additional NODE-based method, STGODE [1], on the river discharge task (Table 1). INDEQS outperforms STGODE by a clear margin (MAE 5.01 ± 0.16 vs. 6.78 ± 0.28; RMSE 16.10 ± 0.65 vs. 17.66 ± 0.34). This situates our method among closely related graph-informed continuous-time approaches and clarifies how it compares to both NCDE- and NODE-style baselines.
>
> [1] Fang et al. "Spatial-Temporal Graph ODE Networks for Traffic Flow Forecasting." KDD 2021.

---

> ### Author Response · Authors · 2026-07-22
> **Author rebuttal II**
>
> > 5. Separate the effect of the informed encoder from the decoder.
>
> To separate encoder and decoder effects, we conducted a matched comparison of graph encoders versus decoders in Appendix A9.4 and Table 9. This analysis contrasts convolutional and continuous decoders across INDEQS_outer and STG-NCDE, helping to disentangle gains due to graph informedness in the encoder from gains attributable to the more expressive decoder. We reference these new results in the main text to ensure that our empirical claims about INDEQS focus on the effect of graph priors rather than decoder choice alone.
>
> We hope these clarifications and additions address the reviewer’s concerns and more accurately communicate the nature of our contribution and the specific regimes in which graph-informed NCDEs provide practical benefits.

---

### Author Response · Authors · 2026-07-22
**Summary of Revisions**

We thank all reviewers for their careful reading and constructive feedback. We have uploaded a revised manuscript; all changes are marked in blue (via latexdiff, please disregard the blue coloring of the reference list, which is an artifact of the diff tool). Below we summarize the main revisions, beginning with points raised by multiple reviewers; detailed point-by-point responses are given in the individual replies.

1. ***New robustness experiment for misspecified graph priors (Reviewers B5Xr, cjFY, pHVx).*** We added an experiment on the graph advection data (Appendix A9.2, Figure 11) in which the ground-truth directed graph is perturbed by removing or adding edges, made undirected, or transposed. INDEQS_outer with a perturbed graph consistently outperforms the uninformed STG-NCDE baseline, and INDEQS_inner does so in all but the transposed case, supporting the claim that partially correct priors remain useful.

2. ***Sharpened positioning and scope (Reviewers B5Xr, pHVx)***. We revised the contribution list at the end of the introduction to state explicitly that INDEQS is a xgraph-informed architectural modification of STG-NCDE. We further clarify in our reply to Reviewer pHVx that we do not claim a fully general framework: INDEQS targets settings where the directed graph approximates the dominant flow of information or mass (e.g., advection, river networks), consistent with the limitations stated in the discussion section.

3. ***Notation, dimensions, and presentation (Reviewers cjFY, pHVx)***. We corrected and harmonized notation across sections, clarified how the tensor operations in each equation are to be understood, rewrote Appendix A.4 with a concrete step-by-step example, added explanatory prose to reduce the formula density of the background and method sections, and corrected the citation formatting throughout.

5. ***Weakened claims on traffic flow (Reviewer B5Xr)***. Following the suggestion to test statistical significance, we performed a Wilcoxon signed-rank test on the pairwise differences between INDEQS_outer and STG-NCDE on PeMS08 over 5 independent runs (Appendix A.8). The test does not indicate a statistically significant improvement (p = 0.563; p = 0.625 for k = 2), despite a consistent positive trend. We have accordingly revised the abstract, introduction, experimental section, discussion, and conclusion: the PeMS08 results are now described as an encouraging but not conclusive indication, and our performance claims focus on the larger, physically structured graphs (advection, river discharge), where the gains are clearer.

7. ***Encoder/decoder ablation (Reviewer B5Xr)***. We added a matched comparison of graph encoders versus decoders (Appendix A9.4, Table 9) to disentangle gains from graph informedness in the encoder from gains attributable to the decoder choice.

6. ***Additional continuous-time baseline (Reviewer B5Xr)***. We added STGODE as a NODE-based graph-informed baseline on the river discharge task (Table 1); INDEQS outperforms it on both MAE and RMSE.

We hope these revisions address the concerns raised, and we would be happy to elaborate on any remaining points during the discussion period.